# Neural Correlates of Serial Dependence: Synaptic Short-term Plasticity Orchestrates Repulsion and Attraction

**Xiuning Zhang[1]**
xiuning_zhang@outlook.com

**Xincheng Lu[1]**
lvxc24@mails.tsinghua.edu.cn

**Nihong Chen[2]\***
nihongch@m.scnu.edu.cn

**Yuanyuan Mi[1]\***
miyuanyuan@tsinghua.edu.cn

## Abstract

Serial dependence reflects how recent sensory history shapes current perception, producing two opposing biases: repulsion, where perception is repelled from recent stimuli, and attraction, where perception is drawn toward them. Repulsion typically occurs at the sensory perception stage, while attraction arises at the post-perception stage. To uncover the neural basis of these effects, we developed a two-layer continuous attractor neural network model incorporating synaptic short-term plasticity (STP). The lower layer, dominated by synaptic depression, models sensory processing and drives repulsion due to sustained neurotransmitter depletion. The higher layer, dominated by synaptic facilitation, models post-perception processing and drives attraction by sustained high neurotransmitter release probability. Our model successfully explains the serial dependence phenomena observed in the visual orientation judgment experiments, highlighting STP as the critical mechanism, with its time constants defining the temporal windows of repulsion and attraction. Furthermore, the model provides a neural foundation for the Bayesian interpretation of serial dependence. This study advances our understanding of how the neural system leverages STP to balance sensitivity in sensory perception with stability in post-perceptual cognition.

## 1 Introduction

The visual world casts dynamic images on the retina. From moment to moment, internal and external noise fluctuates across changes in lighting, occlusion, and viewpoint in the environment. Yet we perceive coherence and stability in the world. Serial dependence is an intrinsic mechanism through which our visual system exploits temporal correlations and contextual redundancies by merging similar stimuli that slightly change over time. The visual system harnesses this autocorrelation by inducing positive serial dependencies, drawing the current perception towards the recent history. Through continuous serial dependencies, cognition compensates for variability in sensory input, stabilizing what would otherwise be a noisy and discontinuous experience of the world [1, 2, 3, 4].

In contrast to the positive serial dependence, negative dependence is widely documented. Examples of negative serial dependence are well-known phenomena of visual adaptation and aftereffects [5, 6], e.g., a vertical grating appears to tilt clockwise after exposure to counterclockwise orientations. Such

---

\*Corresponding authors. [1]Department of Psychological and Cognitive Sciences, Tsinghua University, Beijing 100084, China. [2]Key Laboratory of Brain, Cognition and Education Sciences, Ministry of Education; School of Psychology, Center for Studies of Psychological Application; Guangdong Key Laboratory of Mental Health and Cognitive Science, Philosophy and Social Science Laboratory of Reading and Development in Children and Adolescents; South China Normal University, Guangzhou 510631, China.

39th Conference on Neural Information Processing Systems (NeurIPS 2025).

repulsive biases have been observed in sensory cortex, as a classical behavioral probe for neural encoding of visual features [7, 8], such as orientation [9, 10], motion [11], and face viewpoint [12].

Given the coexistence of attractive and repulsive biases, the issue has been raised about the stage at which the two types of serial dependence originate. There has been growing evidence suggesting that repulsion typically occurs in the sensory stage, while attraction appears in the post-perception stage [13, 14, 15, 16, 17].

This study investigates how the cascade of neural processes is calibrated to the history of sensory and decisional events in neural networks, leveraging the unique characteristics of short-term synaptic plasticity (STP) [24, 25, 26, 27, 28, 29], wherein synaptic efficacy varies dynamically with presynaptic activity. Two competing effects exist: short-term facilitation (STF) and short-term depression (STD). Previous studies have demonstrated that STP can regulate neural information processing effectively [30, 31, 32]. For instance, STD can reduce output correlation by inhibiting high-frequency inputs, thereby reducing the autocorrelation of temporal sequences [33]; STF can facilitate the maintenance of input information by strengthening neuronal connections, thereby promoting the integration of temporal sequences [29]. Postsynaptic neurons receiving STD-dominated or STF-dominated synaptic inputs exhibit low-pass or high-pass filtering responses, respectively, achieving diversified processing of temporal sequences [34]. We hypothesize that STP underpins the neural basis of serial dependence, orchestrating its dynamic perceptual effects.

In this work, we develop a two-layer network model incorporating STP to elucidate the underlying mechanisms for both the attractive and repulsive effects in serial dependence. Specifically, to model the visual orientation judgment experiment, we adopt continuous attractor neural networks (CANNs) for each layer [35, 36]. The lower layer, characterized by STD-dominance, models sensory processing, which produces repulsion via sustained neurotransmitter depletion. The higher layer, characterized by STF-dominance, models post-perception processing, which produces attraction via sustained high neurotransmitter release probability. Our model successfully explains the serial dependence phenomena observed in experiments, revealing that the STP time constants determine the temporal windows of repulsion and attraction. Additionally, our model provides a neural mechanistic explanation of the Bayesian interpretation of serial dependence.

**Related Works** Contemporary models of serial dependence employ efficient encoding and Bayesian decoding to explain perceptual biases [13, 15, 18, 19, 20, 21]. In the encoding phase, prior stimuli recalibrate neural sensitivity, reshaping the probability distribution of the current stimulus to induce repulsive bias. In the decoding phase, Bayesian inference integrates sensory likelihood with a prior, yielding the posterior perception and producing attractive or repulsive biases toward prior stimuli. However, the neural basis of this dual process lacks biological validation, and the underlying circuitry dynamics remain largely unexplored. Moreover, existing models fail to capture the temporal dynamics of serial dependence, such as bias reduction with longer inter-stimulus intervals (ISI). Bridging the Bayesian framework with a mechanistic neural model remains a challenge. Recent neuroimaging advances have spurred exploration of the neural computations underlying serial dependence. Some studies attribute attraction to short-term facilitation (STF) or slow excitatory NMDA currents in higher cortical regions, with enhanced synaptic efficacy or neuronal activity [22, 23]. However, these neural dynamic models cannot explain the repulsion effect and the dynamic balance between attraction and repulsion observed in experiments.

## 2 The Network Model

To unveil the neural mechanism underlying serial dependence, we developed a two-layer neural network with hetero-synaptic STP. Specifically, to model visual orientation processing, we adopted a continuous attractor neural network (CANN) for each layer (Fig. 1A). CANNs are a canonical model for neural information representation [37, 38, 39], which effectively mimics the encoding of continuous features in local circuits, such as visual orientation [40, 41] and spatial location [42, 43, 44]. In the brain, stimulus information propagates from lower to higher cortical areas, establishing hierarchical processing from sensory perception to post-perceptual cognition. STP (Fig. 1B-C) is a fundamental neurophysiological property that dynamically modulates synaptic efficacy based on presynaptic firing history [45, 46], manifesting as either STF or STD. Electrophysiological studies suggest potential distinct regional specialization: sensory cortex exhibit STD-dominance [47, 48, 49, 50], while high-level cortex such as prefrontal cortex (PFC) exhibit STF-dominance [24, 53].

Therefore, the lower layer models STD-dominated processing in the visual cortex (e.g., V1), while the higher layer models STF-dominated processing in high-level cortex (e.g., PFC).

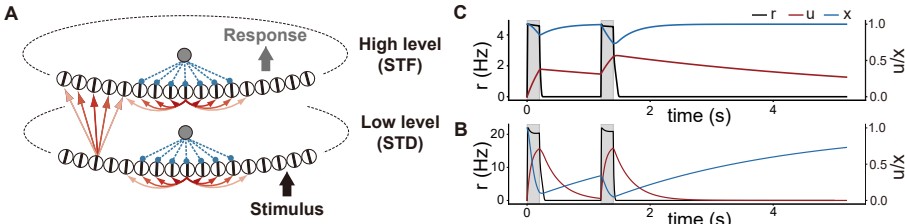

Figure 1: A two-layer CANN with heterosynaptic STP. (A) Schematic of the network model. The hierarchical two-layer architecture mimics STD-dominated processing in the visual cortex and STF-dominated processing in the higher post-perceptual cortex. Excitatory neurons within each layer are aligned in a one-dimensional ring according to their preferred orientation $\theta$, subject to global inhibition (gray solid circle). Intra-layer and inter-layer neurons are connected with color-scaled lines that denote synaptic connectivity strength. (B) An STD-dominated network displays adaptive synaptic efficacy that sustains at low levels. During stimulation (shaded period), neural activity ($r$) increases the neurotransmitter release probability ($u$) while depleting the available neurotransmitter concentration ($x$). Following stimulus offset, $u$ and $x$ recover to baselines. (C) An STF-dominated network retains elevated synaptic efficacy (large value of $u$) for prolonged time.

In each layer of the CANN, neurons are aligned on a one-dimensional ring according to their preferred visual orientations $\theta \in (-\pi/2, +\pi/2]$ (Fig. 1A). Denote $h_M(\theta_M, t)$ the synaptic current to neurons at $\theta_M$ at time $t$, where $M = \{L, H\}$ indexes the lower (L) or higher (H) layer, and $r_M(\theta_M, t)$ the corresponding firing rate. The neuronal dynamics are described as:

$$\tau_L \frac{\partial h_L(\theta_L, t)}{\partial t} = -h_L(\theta_L, t) + \rho \int J_{LL}(\theta_L, \theta'_L) u_L(\theta'_L, t) x_L(\theta'_L, t) r_L(\theta'_L, t) d\theta'_L \tag{1}$$
$$+ I_{ext}(\theta_L, t) + \mu_L \xi_b(\theta_L, t),$$

$$\tau_H \frac{\partial h_H(\theta_H, t)}{\partial t} = -h_H(\theta_H, t) + \rho \int J_{HH}(\theta_H, \theta'_H) u_H(\theta'_H, t) x_H(\theta'_H, t) r_H(\theta'_H, t) d\theta'_H$$
$$+ \int J_{HL}(\theta_H, \theta'_L) r_L(\theta'_L, t) d\theta'_L + \mu_H \xi_b(\theta_H, t), \tag{2}$$

where $\tau_M$, with $M = \{L, H\}$, denotes the time constant of neurons in the lower or higher layer, respectively. $\rho$ is the neuronal density. $\xi_b(\theta_M, t)$ denotes background Gaussian white noises of zero mean and unit variance, and $\mu_M$ the noise strength. The firing rate of neurons is calculated by, $r_M(\theta_M, t) = h_M^2(\theta_M, t) / [1 + k_M \rho \int h_M^2(\theta'_M, t) d\theta'_M]$, with the parameter $k_M$ controlling the divisive normalization strength [39]. We set the neuronal connections in the same layer or between layers to be $J_{KM}(\theta_K, \theta'_M) = J_{KM}^0/(\sqrt{2\pi} a_{KM}) \exp\left[-(\theta_K - \theta'_M)^2/(2a_{KM}^2)\right]$, with $K, M \in \{L, H\}$), where $J_{KM}^0$ denotes the maximum strengths for intra-layer connections ($J_{LL}^0, J_{HH}^0$) and inter-layer connections ($J_{HL}^0$), and $a_{KM}$ controls the neuronal interaction range. Importantly, $J_{KM}$ depends only on the difference ($\theta_K - \theta'_M$), which is translation-invariant in the feature space, a crucial property enabling a CANN to maintain a continuum of attractors to represent a continuous feature [39, 51, 52].

STP modulates the synaptic efficiency between neurons, which is characterized by two variables, the neurotransmitter release probability $u$ and the available neurotransmitter resource $x$. When a presynaptic neuron fires, the calcium accumulation at its axon terminal triggers two competing processes: 1) STF, due to the increase of neurotransmitter release probability $u$ and 2) STD, due to depletion of $x$. The instantaneous synaptic efficacy is $ux$. The dynamics of STP are given by:

$$\frac{\partial u_M(\theta_M, t)}{\partial t} = -\frac{u_M(\theta_M, t)}{\tau_f^M} + U_0^M (1 - u_M(\theta_M, t)) r_M(\theta_M, t), \quad M = L, H \tag{3}$$

$$\frac{\partial x_M(\theta_M, t)}{\partial t} = \frac{1 - x_M(\theta_M, t)}{\tau_d^M} + u_M(\theta_M, t) x_M(\theta_M, t) r_M(\theta_M, t), \quad M = L, H \tag{4}$$

where the time constants $\tau_f$ and $\tau_d$ determine how quickly $u$ and $x$ recover to baselines, respectively. Since a larger $\tau_f$ implies that $u$ remains at high-level after neuronal response for a longer time, it

corresponds to the STF effect; while a larger $\tau_d$ implies that $x$ remains at low-level for a longer time, it corresponds to the STD effect. Experimental data show that sensory cortices like V1 are STD-dominated, i.e., $\tau_d \gg \tau_f$ [47], while higher cortical regions such as PFC are STF-dominated, i.e., $\tau_f \gg \tau_d$ [24, 53]. We set STP parameters in the lower and higher layers of our model accordingly (Fig. 1B-C), and the time constants also match the timescales of working memory [29, 31].

## 3 STP Induces Repulsion and Attraction in a Single-Layer CANN

Before studying the performance of the two-layer network, we first explore how STP induces serial-dependent biases in a single-layer CANN.

To compare the network performance with experimental data, we adopted the post-cueing adjustment paradigm (Fig. 2) [54, 55, 56, 57], where participants viewed two sequentially presented visual stimuli $S_1$ and $S_2$, whose orientations $\theta_1^s$ and $\theta_2^s$ are randomly sampled in the range of $(-\pi/2, \pi/2]$ in each trial. After a delay period, participants reported the memorized orientation, denoted as $\theta_{\text{cue}}^d$, giving the cuing signal (Fig. 2A, D top). External visual stimuli and the cueing signal are presented to the lower layer, denoted as $I_{\text{ext}}(\theta_L, t)$, with $\text{ext} \in \{\text{sti}, \text{cue}\}$, which are expressed as $I_{\text{ext}}(\theta_L, t) = \alpha_{\text{ext}} \exp\left[-(\theta_L - \theta_{\text{ext}})^2/(2a_{\text{ext}}^2)\right] + \mu_{\text{ext}}\xi_{\text{ext}}(\theta_L, t)$, with $\alpha_{\text{ext}}$ controlling the signal strength, $a_{\text{ext}}$ the signal width (inversely related to the precision), and $\mu_{\text{ext}}$ the noise strength. In the retrieval phase, we modeled the cueing signal as a weak and ambiguous copy of the corresponding stimulus signal by setting the parameters $\alpha_{\text{cue}} \ll \alpha_{\text{sti}}$, $a_{\text{cue}} > a_{\text{sti}}$, and $\mu_{\text{cue}} > \mu_{\text{sti}}$ [32], with detailed noise settings in Appendix A. We manipulated the variables, including the inter-stimulus interval (ISI, $\Delta t_{\text{ISI}}$) and the inter-trial interval (ITI, $\Delta t_{\text{ITI}}$), and measured the adjustment error of the model, which is given by $\text{Error} = \theta_2^d - \theta_2^s$, and compared it to the stimulus orientation difference $\Delta S = \theta_1^s - \theta_2^s$.

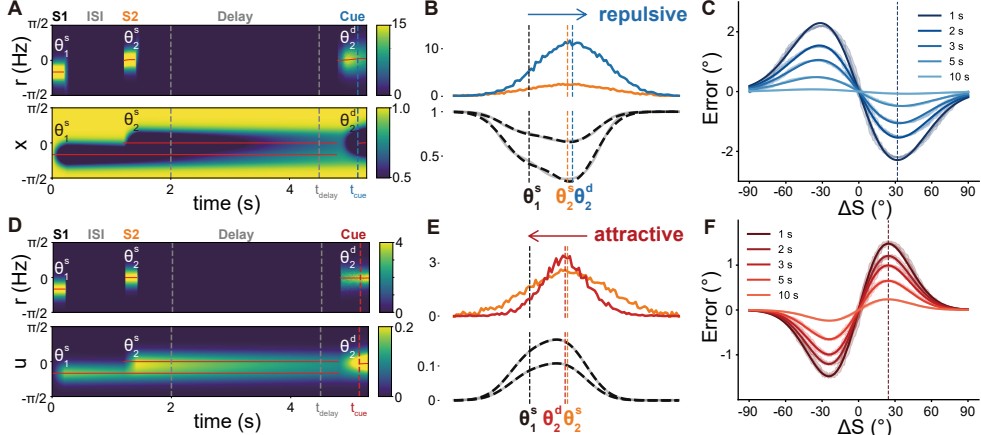

Figure 2: Serial dependence in a single-layer CANN with STP. (A-C) Repulsion effect induced by STD-dominance. (D-F) Attraction effect induced by STF-dominance. (A, D) Schematic illustration of the temporal dynamics of neural responses $r$, the available neurotransmitter resource $x$, and the neurotransmitter release probability $u$ in a single-layer CANN with STD-dominance (A) or STF-dominance (D). Two stimuli $S_1$ and $S_2$ with respective orientations $\theta_1^s = -30°$ and $\theta_2^s = 0°$ are presented sequentially. A cueing signal is presented after the delay period to trigger the retrieval of $S_2$. (Top) The red line marks the decoded stimulus orientation. (Bottom) The red lines during stimulus/delay periods denote stimulus orientations in current trial. The recall-period red line indicates the recalled orientation of the second stimulus. (B, E) Illustration of the repulsive and attractive biases. In the STD-dominated case (B), the retrieved neural response ($\theta_2^d$, blue curve) is repelled away from the earlier presented stimulus ($\theta_1^s$). In the STF-dominated case (E), the retrieved neural response is attracted toward $\theta_1^s$. These biased effects exist in both the neural response (top) and in the neurotransmitter concentration (bottom). Gray and dotted curves represent simulation and fitting results, respectively. (C, F) Adjustment error (Error $= \theta_2^d - \theta_2^s$) as a function of the difference between two stimuli ($\Delta S = \theta_1^s - \theta_2^s$) under different ISI conditions. It displays negative correlations (repulsion) in the STD-dominated case (C) and positive correlations (attraction) in the STF-dominated case (F). The shaded area represents standard error across simulation runs. For more details, see Appendix A.

## 3.1 STD-dominance induces repulsion in a single-layer CANN

We first demonstrated that a single-layer CANN with STD-dominance induces serial-dependent repulsion. In each trial, stimuli $S_1$ and $S_2$ activate neurons sequentially in the form of bump-shaped activities centered at $\theta_1^s$ and $\theta_2^s$, respectively (Fig. 2A top), which deplete synaptic resources of active neurons (Fig. 2A bottom) and reduce their synaptic efficacy. The neurotransmitter concentration $x(\theta, t)$ depleted by $S_1$ and $S_2$ forms a double-trough distribution with minima at $\theta_1^s$ and $\theta_2^s$ (red curves in Fig. 2A bottom). The late presentation of $S_2$ implies the late recovery of corresponding neurotransmitters, resulting in $x(\theta_2^s, t) < x(\theta_1^s, t)$ (gray curve in Fig. 2B, bottom). Due to the time constants $\tau_d \gg \tau_f$, neurotransmitter concentration remains at low levels during the maintenance period, and they can be mathematically expressed as, $x(\theta, t) = 1 - A_x^1(t) \exp\left[-(\theta - \theta_1^s)^2/2a^2\right] - A_x^2(t) \exp\left[-(\theta - \theta_2^s)^2/2a^2\right]$, validated by our numerical simulation (dotted gray curve in Fig. 2B, Appendix C; the variation of $u(\theta, t)$ and $u(\theta, t)x(\theta, t)$ over time, see Appendix D). When the retrieval cue for $S_2$ is presented, the network generates a response bump (blue curve in Fig. 2B, top) centered at $\theta_2^d$ (decoded via population vector from neural activity, Appendix A). Notably, before presenting the retrieval cue, the double-trough distribution of $x(\theta, t)$ is lower on the side of $\theta_2^s$ closer to $\theta_1^s$, resulting in weaker synaptic connections in this region. This creates asymmetric neuron interactions, pushing the network response $\theta_2^d$ away from $\theta_1^s$, manifesting the repulsion effect (Fig. 2B).

We conducted 20 simulation runs (100 trials each, cue = 2 as examples), modeling behaviors of different participants (Appendix A). Analysis using Derivative of Gaussian function (DoG, $G(x) = \sqrt{e}(x/\sigma)A_{\text{DoG}} \exp(-x^2/2\sigma^2)$, with $A_{\text{DoG}}$ denoting the curve's amplitude, Fig. 2C) revealed a negative correlation between the stimulus similarity $\Delta S$ and the adjustment error Error (one-sample t-test against zero on the sum of error: t(19) = 197.36, p < .001). Stimuli with higher similarity (smaller $\Delta S$) generate stronger repulsion (larger $|\theta_2^d - \theta_2^s|$). We found that the amplitude $A_{\text{DoG}} = -2.29°$ and the repulsion effect peaks at $\Delta S = 31.71°$, in agreement with the V1 repulsion observed in psychophysical experiments[15, 16, 17]. To test generality, we re-ran the one-layer model by randomly cueing $S_1$ or $S_2$ in interleaved trials. The results showed that the adjustment error curve aligns with that in Fig. 2C, with a magnitude of -3.51° (t(19) = 168.57, p < .001, Appendix E).

By manipulating ISI, we further investigated the regulation of STD on the repulsion effect, and found that: when $\Delta t_{\text{ISI}} < \tau_d$, decreasing $\Delta t_{\text{ISI}}$ increases the repulsion effect (all t(19) > 43.18, p < .001). Conversely, if neurotransmitters are sufficiently recovered, the repulsion is reduced significantly.

## 3.2 STF-dominance induces attraction in a single-layer CANN

We continued to explore how STF affects serial dependent biases in a single-layer CANN. As shown in Fig. 2D, two sequentially presented stimuli $S_1$ and $S_2$ trigger neural activity bumps centered at $\theta_1^s$ and $\theta_2^s$, respectively, causing the increase of neurotransmitter release probability of active neurons, and so do the interactions between them. Since the neural activity bumps caused by two stimuli are superimposed, creating a bimodal distribution of $u(\theta, t)$ peaked at $\theta_1^s$ and $\theta_2^s$. The latter-presented $S_2$ induces a slower decay of the corresponding neurotransmitter release probabilities, resulting in $u(\theta_2^s, t) > u(\theta_1^s, t)$. Since the time constant of STP $\tau_f \gg \tau_d$, the neurotransmitter release probabilities remain at high levels during the maintenance period, which can be mathematically expressed as $u(\theta, t) = A_u^1(t) \exp\left[-(\theta - \theta_1^s)^2/2a^2\right] + A_u^2(t) \exp\left[-(\theta - \theta_2^s)^2/2a^2\right]$, validated by numerical simulation (dotted gray curve in Fig. 2E bottom, Appendix C; the variations of $x(\theta, t)$ and $u(\theta, t)x(\theta, t)$ over time see Appendix D). During the retrieval phase (cue = 2 as examples), the neurotransmitter release probabilities near $\theta_2^s$ on the side closer to $\theta_1^s$ have larger values than those on the side away from $\theta_1^s$, causing stronger neuronal interactions in this region. Consequently, this shifts the network response center $\theta_2^d$ triggered by the cueing signal towards $\theta_1^s$, manifesting the attraction effect (Fig. 2E top). We found that the adjustment error (Error) correlates positively with the stimulus similarity (t(19) = 122.92, p < .001), with a DoG amplitude $A_{\text{DoG}} = 1.48°$, peaking at a $\Delta S = 24.52°$ (Fig. 2F), which is in agreement with the attraction effect observed in PFC and other higher cortical areas[17, 60, 61]. When randomly cueing $S_1$ or $S_2$, the adjustment error curve aligns with that in Fig. 2F, with a magnitude of 1.68° (t(19) = 96.97, p < .001, Appendix E).

By varying ISI (Fig. 2F), we found that the attraction effect ($A_{\text{DoG}}$) increases as $\Delta t_{\text{ISI}}$ decreases, under the condition $\Delta t_{\text{ISI}} < \tau_f$ (all t(19) > 85.63, p < .001). Conversely, when neurotransmitter release probabilities decay to the baseline, the attraction effect is reduced significantly.

# 4 STP Orchestrates Repulsion and Attraction in a Two-layer Network

## 4.1 Reproducing perceptual biases in a visual orientation judgment task

Using the post-cueing paradigm, we further demonstrated that heterosynaptic STP in two layers of the model induces repulsive and attractive effects, respectively. We carried out 20 runs, each containing 100 consecutive trials. Let us consider the $(n+1)$th trial (Fig. 3A), two visual stimuli $S_i^{n+1}$, with orientation $\theta_i^{s,n+1}$, for $i = 1, 2$, are presented to the lower layer of the network sequentially, which trigger neuronal responses in the bump-shape centered at $\theta_{i,M}^{s,n+1}$ ($M = L, H$), respectively, in each layer. We implemented a recall paradigm, in which a randomly selected stimulus serves as the recall cue ($S_{\text{cue}}^{s,n+1}$, cue $= \{1, 2\}$) and triggers a retrieved bump activity centered at $\theta_{\text{cue},M}^{d,n+1}$. We then calculated the adjustment errors (Error $= \theta_{\text{cue},H}^{d,n+1} - \theta_{\text{cue}}^{s,n+1}$) for cued stimuli and their relationship to both the orientation difference in the current ($\Delta S_{\text{within}} = \theta_{\text{uncue}}^{s,n+1} - \theta_{\text{cue}}^{s,n+1}$, **within-trial**) and the orientation difference cued in the preceding trial ($\Delta S_{\text{between}} = \theta_{\text{cue}}^{s,n} - \theta_{\text{cue}}^{s,n+1}$, **between-trial**).

**Within-trial repulsion.** According to the study in Sec. 3.1, in the lower-layer CANN with STD-dominance (modeling information processing in V1), $\theta_{\text{cue},L}^{d,n}$ in any $n$th trial exhibits the repulsion effect from other within-trial stimuli, as shown by the orange curve in Fig. 3B (bottom). This effect arises due to the spatially asymmetric distribution of available neurotransmitters ($x_L(\theta_L, t)$) caused by STD (gray curve). Notably, while disturbances in $x_L(\theta_L, t)$ from previous trial stimuli ($S_i^{n-1}$) persist, their influence is negligible because $\tau_d^L < 2T_{\text{trial}}$ ($T_{\text{trial}}$ is the trial duration), and that $\theta_i^{s,n-1}$ and $\theta_{\text{cue},L}^{d,n-1}$ are uniformly distributed around $\theta_i^{s,n}$, making the cumulative effect statistically insignificant during the $n$th trial delay. The repulsively shifted neuronal responses in the lower layer are transmitted to the higher layer via feedforward connections (black curve in Fig. 3B top). Different from the lower layer, because of STF-dominance, the high layer has enhanced neurotransmitter release probabilities $u_H(\theta_H, t)$ in the region around the current trial stimuli (gray curve in Fig. 3B top). This enhanced synaptic efficacy selectively amplifies feedforward inputs centered at $\theta_{\text{cue},L}^{d,n}$. These repulsively shifted inputs from the low layer make the triggered neural response in the high layer exhibit repulsion (orange curve in Fig. 3B top).

**Between-trial attraction.** After two visual stimuli in the $(n+1)$th trial are presented, neurotransmitter release probabilities $u_H(\theta_H, t)$ in the higher layer exhibit a three-peak distribution, peaking at $\theta_1^{s,n+1}$, $\theta_2^{s,n+1}$, and $\theta_{\text{cue,H}}^{d,n}$ (Appendix C). Since $u_H(\theta_H, t)$ decays slowly in the high-layer, it maintains at a relatively high level during the delay period of the $n$th trial, and the neuronal activity induced by the retrieval cue in the high-layer further facilitates the neurotransmitter release probabilities around $\theta_{\text{cue},H}^{d,n}$ (Fig. 3A bottom), resulting in an amplitude hierarchy during the delay period in the $(n+1)$th trial: $A_u^{\text{cue},n}(t) \gg A_u^{1/2,n+1}(t)$ (gray curve in Fig. 3C top). This dynamic characteristic causes the neural activity induced by the recall cue in the $(n+1)$th trial to shift toward $\theta_{\text{cue},H}^{d,n}$, manifesting the attraction effect (orange curve in Fig. 3C top).

Statistical analysis shows that the model's adjustment error (Error $= \theta_{\text{cue},H}^{d,n+1} - \theta_{\text{cue}}^{s,n+1}$) negatively correlates with the within-trial stimulus similarity ($\Delta S_{\text{within}}$), peaking at $38.13°$ ($A_{\text{DoG}} = -0.91°$, t(19) = 12.62, p < .001; Fig. 3D, blue curve); whereas, the network's adjustment error positively correlates with the between-trial stimulus similarity ($\Delta S_{\text{between}}$), peaking at $17.83°$ ($A_{\text{DoG}} = 1.18°$, t(19) = 7.94, p < .001; Fig. 3D, red curve). Notably, these two opposite effects are consistent with the repulsive and attractive biases observed in post-cue behavioral and neurophysiological experiments[58, 59, 61], and their magnitudes are comparable to the strength of classical serial dependence effects reported in the literature (e.g., $A_{\text{DoG}} = 1.17°$[14], $A_{\text{DoG}} = 1.32°$[20], $A_{\text{DoG}} = 1.59°$[59]). To account for potential readout bias due to asymmetric neural responses, we decoded high-layer activity using multiple methods (population vector method, center-of-mass, maximum likelihood, and peak decoding, Appendix F). All approaches yielded consistent results, confirming robust within-trial repulsion and between-trial attraction.

Psychophysical experiments also indicate that reporting the cued stimulus in the previous trial significantly affects the current between-trial attraction. By applying a partial no-report paradigm (see Appendix B), we found that when no report is required in the previous trial, the current target is less attracted to the previous one, and the within-trial repulsion effect increases slightly (between:

$A_{\text{DoG}} = 0.14°$, t(19) = 0.26, p = .802; within: $A_{\text{DoG}} = -1.29°$, t(19) = 9.21, p < .001; Fig. 3E), consistent with the psychophysical experimental data [1, 62]. Based on this, our model further predicts that compared to actual visual stimulus values (i.e., $\Delta S_{\text{between}} = \theta_{\text{cue}}^{s,n} - \theta_{\text{cue}}^{s,n+1}$), the retrieved stimulus contributes more to the between-trial attractive effect ($A_{\text{DoG}} = 1.28°$, t(19) = 8.9, p < .001; Fig. 3F), which is consistent with the psychophysical experimental data [20]. Thus, the between-trial attraction effect stems more from the memory representation difference (i.e., $\Delta R_{\text{between}} = \theta_{\text{cue,H}}^{d,n} - \theta_{\text{cue}}^{s,n+1}$).

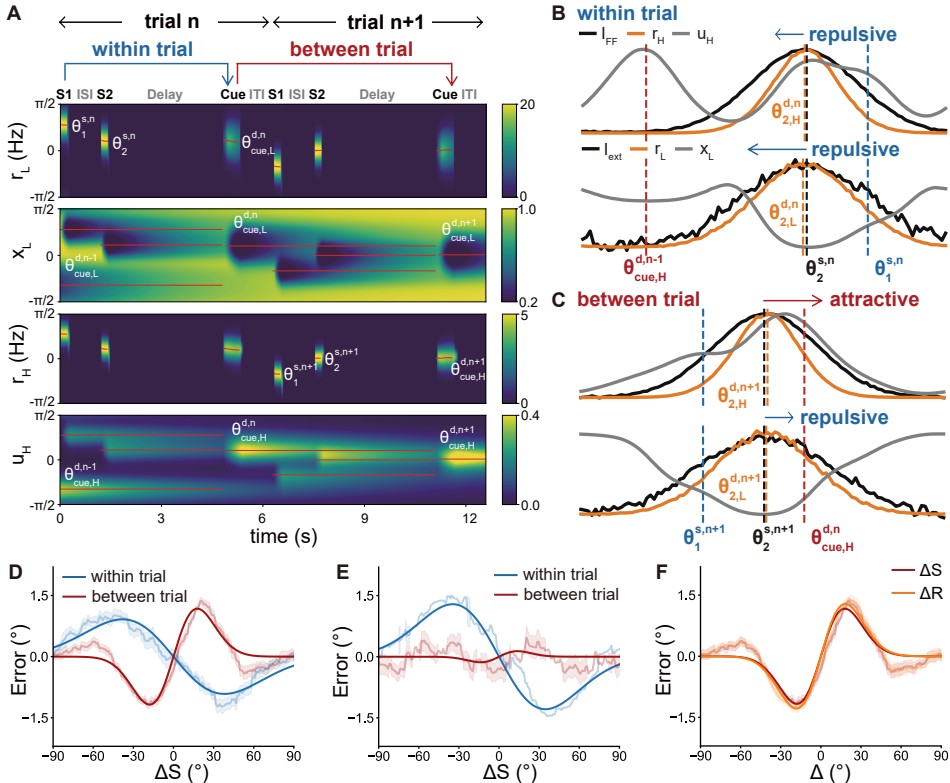

Figure 3: Serial dependence in the two-layer CANN model. (A) Temporal dynamics of the model variables. Two stimuli presented sequentially evoke neural responses. Post-delay, the network retrieves $S1$ or $S2$ based on the task cue. Plots (top to bottom): lower-layer firing rate, lower-layer available neurotransmitter resource, higher-layer firing rate, and higher-layer neurotransmitter release probability. After stimulus offset, $r_L(\theta_L)$ and $r_H(\theta_H)$ return to baseline; $x_L(\theta_L)$ and $u_H(\theta_H)$ persist, affecting decoding within and across trials. (B) Within-trial repulsion in the two-layer model. Low-layer decoded signal ($\theta_{2,L}^{d,n}$, orange curve) is biased away from the non-target ($\theta_1^{s,n}$) due to the asymmetric neurotransmitter concentration profile (gray). This propagates to the higher layer, where spatial asymmetry in $u_H(\theta_H, t)$ (gray) shifts the response bump, causing repulsion. (C) Between-trial attraction in the two-layer model. Lower-layer signal propagates to the higher layer, where $u_H(\theta_H, t)$ asymmetry (gray, top) in (n+1)th trial shifts the response bump (orange curve, top) toward $\theta_{\text{cue,H}}^{d,n}$, causing attraction. All variables normalized. (D) Within-trial repulsion and between-trial attraction. Retrieval error (angular difference between higher-layer readout and cued stimulus) plotted against the difference between the prior and current stimuli. (E) Within-trial and between-trial biases without cueing in the prior trial. (F) Within-trial and between-trial biases plotted against the difference between the prior stimulus (or its report) and the current stimulus. For more details, see Appendix B.

## 4.2 The neural basis of Bayesian interpretation of serial dependence

Contemporary models in the field explaining repulsion and attraction in serial dependence often adopt the Bayesian inference framework, such as the gain model [1] and two-process model [13, 20]. These models consider the whole process involving two stages (Fig. 4A): 1) sensory encoding, in which orientation-selective neurons in the sensory cortex encode stimuli in the form of tuning

curves, i.e., $r(\theta_k) = \alpha \exp\left[\beta \cos(\theta - \theta_k) - 1\right]$, with $\theta_k$ denoting the orientation preference and $\alpha$, $\beta$ controlling the amplitude and width of the tuning curve, respectively. 2) decision integration, in which higher cortical areas estimate stimuli through weighted integration, i.e., $\theta^d = \sum_{\theta_k} \vec{r}(\theta_k) w_k$, with $w_k$ representing the decision weight. Although differing in regulatory strategies, both the gain and two-process models posit that the brain modulates these two processes during sequential information processing. In particular, the two-process model assumes that: 1) at the sensory encoding stage, neurons undergo adaptive sensitivity regulation, reducing tuning curve amplitudes near previous stimuli $\theta^s$ (black curves in Fig. 4A, modifying $\alpha$ based on $(\theta_k - \theta^s)$), which effectively alters the likelihood function in Bayesian inference; 2) at the decision stage, higher cortical areas develop weight biases toward previous stimuli (green curves in Fig. 4A, modifying $w_j$ based on $(\theta_k - \theta^s)$), equivalent to changing the prior distribution in Bayesian inference. These dual operations shift the posterior distribution, producing repulsion in sensory perception and attraction in post-perception, but the neural basis of these operations is unknown. Also, the existing models have not explained the dynamic characteristics of serial dependence, such as the temporal decay of serial-dependent effects, the within-trial repulsion and between-trial attraction.

Our two-layer CANN model with hetero-STP provides a neural basis for the Bayesian interpretation of serial dependence, specifically, 1) negative regulation at sensory encoding: neurons responding to the presented orientation $\theta_i^s$ will experience resource depletion after activation (Fig. 4B bottom, black curve, equation see Appendix C). This reduces neuronal synaptic efficacy and affects the feedforward transmission to the higher layer, analogous to the modulation of the likelihood function in the Bayesian framework. It drives the repulsion effect as we have analyzed (Fig. 4B bottom, blue curves). 2) positive regulation at the decision stage: neurons in the higher layer responding to the previous stimulus have enhanced neurotransmitter release probabilities to the cueing orientation (Fig. 4B, green curve, Appendix C). This creates biased strong neuronal connections in the network toward the stimulus history, analogous to the modulation of the prior in the Bayesian framework. When a new stimulus arrives (the likelihood information), the high-layer CANN whose connections are implicitly modulated by the stimulus history (the prior) effectively carries out the history-dependent computation (the posterior).

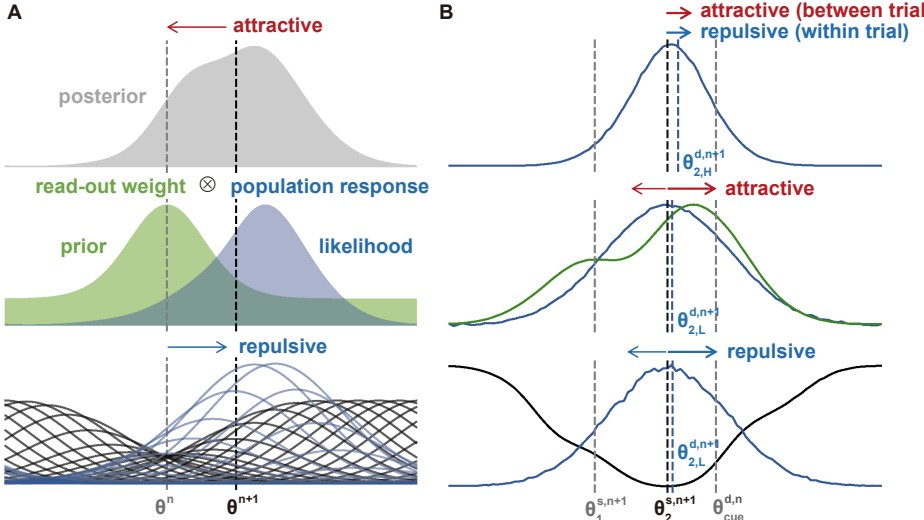

Figure 4: An STP-based Bayesian interpretation of serial dependence. (A) Schematic of Bayesian interpretation. Orientation stimuli (black curves) generate neural population responses (blue curves, likelihood), integrated by readout weights (green distributions) to produce the posterior decision. Adaptation induces repulsive effects; recent stimulus history amplifies decisional weights, eliciting attractive effects. (B) STP correlations of Bayesian inference. Bottom: neurons in the lower layer respond to the prior cued signal and current stimuli, inducing neurotransmitter depletion (gray curve). Recent stimuli evoke stronger repulsion, resulting in a net repulsive bias in the population response (blue curve). Middle: Lower-layer response (blue curve, likelihood) propagates to the higher-layer network; neurotransmitter release probability (green curve, prior) biases toward the cued target. Top: Higher-layer population response (blue curve) displays repulsive bias from the stimulus from the same trial and attractive bias toward the prior cued target.

# 5 Model Prediction

Within-trial repulsion effects reflect neural network sensitivity to rapid changes, while between-trial attraction effects demonstrate stability in long-term processing. In our model, these are due to the fact that STD reduces neurotransmitters, while STF increases neurotransmitter release probabilities. STP, with its time constants positioned between rapid neural encoding (hundreds of milliseconds) and experiential learning (seconds), serves as the suitable neural correlate of adaptive cognitive functions, including motor control, speech recognition, and working memory. In our two-layer network, heterogeneous STP regulates the different information processing in two layers, specifically, the lower layer employs STD for information separation, while the higher layer employs STF for information integration. This area-specific STP regulation provides a way for the neural system to balance information separation and integration, where the STP time constants determine the time boundaries for information separation and integration in temporal sequence processing.

To verify our hypothesis on the role of STP, we used the post-cueing paradigm to study the serial dependence effects by manipulating ISI and ITI time windows accordingly. We conducted 20 runs (100 trials each) for various parameter conditions. The relationship between the judgment error and the stimulus difference was fitted using a DoG curve, with amplitude ($A_{\text{DoG}}$) derived from this fitting (hollow circles in Fig. 5). We have two key observations: (i) When ISI within a trial is shorter than the lower layer's STD time constant ($\tau_d$), the network shows a significant repulsion effect in judgment error (0s: t(19) = 8.66, p < .001), with its strength inversely proportional to ISI. When ISI exceeds $\tau_d$, repulsion transitions to attraction (5s: t(19) = 4.32, p < .01; 10s: t(19) = 5.42, p < .001). Thus, the STD time constant determines the time window of repulsion, as illustrated in Fig. 5A. (ii) When ITI is shorter than the higher layer's STF time constant $\tau_f$, the network exhibits a significant attraction effect in judgment error (all t(19) > 4.84, p < .001), with its strength inversely proportional to ITI. As ITI exceeds $\tau_f$, the attraction effect weakens substantially and becomes negligible when ITI is sufficiently long (both p > .05). Thus, the STF time constant determines the time window of attraction, as shown in Fig. 5B.

The cross-talk between two layers in our model is at the neural activity level, which generates different serial dependence biases. The bottom-up repulsion competes with the STF-maintained prior information (attractive bias), yielding the observed behavioral pattern—attraction within trials and repulsion between trials. This cross-layer competition exhibits time dependency: psychophysical evidence indicates that the attractive effect is strengthened with longer delays[14], reflecting faster decay of STD-driven repulsion versus STF-sustained attraction. Our model (Fig. 5) captures this time-dependent interaction, where the short timescale of STD leads to a faster decay of repulsion, while STF in the higher layer maintains the attractive bias over longer duration.

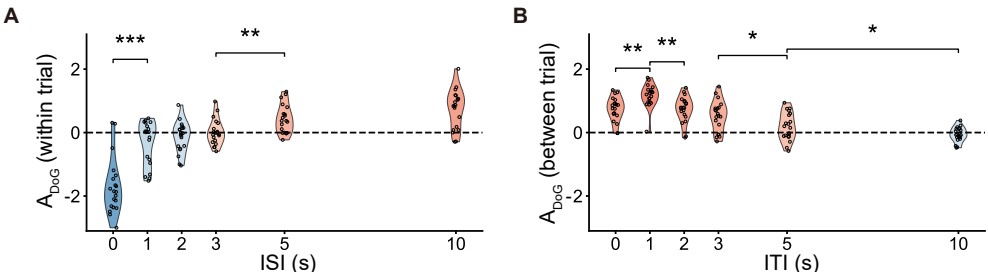

Figure 5: Model predictions on temporal windows of information segregation and integration. (A) The adjustment error (amplitude of the DoG) reveals within-trial repulsion, which gradually decreases and reverses to attraction as the ISI increases. (B) The adjustment error reveals between-trial attraction, which decreases as the ITI increases. *: p < .05; **: p < .01; ***: p < .001.

# 6 Discussion

This study proposes a hierarchical computational strategy for the neural system in dynamic sequential processing, where the STP time constants critically determine the temporal windows and boundaries for information segregation and integration. By incorporating region-specific STP heterogeneity, our continuous attractor neural networks advance the synaptic theory of working memory [29], which

posits that presynaptic neurotransmitters serve as a dynamic buffer—loaded, refreshed, and read out by spiking activity to enable both activity-based and activity-silent memory representations. In V1, STD-dominated plasticity induces a transient repulsive effect via neurotransmitter depletion, driving stimulus-specific suppression, in reminiscence of the classical visual adaptation effect [18, 47, 63, 64, 65]. In contrast, STF-dominance in PFC sustains synaptic efficacy via calcium accumulation, thereby generating attractor dynamics that support temporal integration for working memory and decision-making [29, 66, 67]. Specifically, the time constants of STD and STF dictate the temporal windows for segregation and integration, respectively, providing testable metrics for future research.

The current model resolves the apparently conflicting effects of serial dependence within a unified framework. Serial dependence, initially described as the attraction of current perception to recent stimuli, has been extended to reveal a dissociation between repulsive effects at the perceptual stage and attractive effects at post-perceptual stages [14, 15, 16, 17, 60, 61]. The neural mechanisms underlying these bidirectional biases have been widely debated, with dual-process models proposing an efficient-coding account in the sensory cortex and a decisional-inertia account in higher-order regions [13, 15, 20, 21]. The repulsive effect reflects an optimization strategy in sensory systems [19], where STD-mediated local inhibition maximizes sensitivity to stimulus changes [69]. Conversely, the attractive effect relies on STF-driven synaptic enhancement preserving historical information to enhance decoding efficiency in statistically regular environments. The order of STD and STF is crucial for the model performance. A reversed configuration—lower STF-dominated layer and higher STD-dominated layer—yielded exclusively attractive biases. Both within-trial (amplitude: 1.79°; t(19)=6.79, p<.001) and between-trial effects (amplitude: 3.81°; t(19)=19.62, p<.001; Appendix G) showed significant attraction. Reversing this order (STF-dominance first) causes initial signal smoothing due to facilitation. Consequently, even though the downstream STD layer attempts to extract differential features, the loss of sensitivity to input variation cannot be recovered, resulting in failure to achieve the intended synergy between sensitivity and stability. Sensitivity analyses confirmed that the key repulsion and attraction effects were robust to parameter variations: changing the absolute values of $\tau_d$ and $\tau_f$ (while maintaining $\tau_d \gg \tau_f$, Appendix H) or introducing ±10% random perturbations to STP parameters (Appendix I) produced only minor amplitude changes. This indicates that the results depend on the relative rather than absolute timescales, demonstrating strong model robustness.

**Limitations and Future Works**    Our model elucidates how STP underlies Bayesian computation in biological networks. Bayesian models, such as dual-process frameworks [13, 15, 20, 21], have long been used to explain serial dependence but often lack biological substrates and fail to account for temporal dynamics. In contrast, the two-layer CANN model with hetero-STP offers a biologically grounded implementation: the likelihood function maps to STD-modulated sensory representations, the prior corresponds to STF-sustained memory traces, and their interaction yields the posterior. Unlike the conventional Bayesian models, which posit priors are shaped by long-term environmental statistics [19], the Bayesian theory of serial dependence considers experience-dependent modulations of likelihood function and prior in short timescale, and our model provides a potential neural mechanism to support this view. Since the Bayesian view of serial dependence has been recognized in the field, we hope that linking our model to this view will strengthen our understanding of the neural mechanism of serial dependence. The theoretical and mathematical foundations of this mapping require further analysis. Our model accounts for most documented serial dependence phenomena and their associated neural mechanisms, yet a few experimentally observed patterns require further investigation. For example, long-timescale repulsive effects (>50 s)[21] suggest sensory cortex mechanisms beyond current STP timescales. Implementing long-term negative feedback mechanisms in sensory cortices may explain this. In addition, future work could incorporate feedback projections to simulate top-down modulation, accounting for attractive effects observed in the sensory cortex [70]. Integrating PFC cognitive control theories could elucidate the role of higher-level attention in information processing [71, 72]. These advances will deepen our understanding of the balance between information integration and segregation, opening up new avenues in brain-inspired computing.

## Acknowledgements

This work was supported by the National Science and Technology Innovation 2030 Major Program (No. 2021ZD0203700 / 2021ZD0203705, Y.Y. Mi; No. 2021ZD0203600, N.H. Chen; No. 2021ZD0204103, H. Luo), NSFC 31930053, grant from Research Center for Brain Cognition and Human Development, Guangdong, China (No. 2024B0303390003), and Science Fund for Creative Research Groups of the National Natural Science Foundation of China (T2421004). We are deeply grateful to Prof. Huan Luo and Dr. Huihui Zhang for their generous advice and insightful discussions throughout the course of this study. Their constructive feedback and encouragement have been invaluable in shaping the ideas and refining the presentation of this work.

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

# Appendix

## A Model Details

**Single-layer CANN and two-layer CANN.** The neural dynamics in a single-layer CANN are described as:

$$\tau\frac{\partial h(\theta,t)}{\partial t} = -h(\theta,t) + \rho \int J(\theta,\theta')u(\theta',t)x(\theta',t)r(\theta',t)d\theta' + I_{\text{ext}}(\theta,t) + \mu_b\xi_b(\theta,t) \quad (5)$$

The calculation of $u(\theta,t)$, $x(\theta,t)$, $r(\theta,t)$ was identical to the two-layer CANN in Sec. 2. The time constants for short-term depression ($\tau_d$) and short-term facilitation ($\tau_f$) were set for the STD-dominated CANN and the STF-dominated CANN models, respectively.

We considered heterogeneity in intra-layer and inter-layer neuronal connections across different participants by including noise: $\widetilde{J}_{KM}(\theta_K,\theta'_M) = J_{KM}(\theta_K,\theta'_M)(1+\mu_J\xi_J)$, with $K,M \in \{L,H\}$. $\xi_J$ denotes white Gaussian noise of zero mean and unit variance and $\mu_J$ the noise strength. We included noise in the neuronal connections of the single-layer CANN similarly. Model parameters are shown in Tab. 1.

**Population vector method.** We decoded orientation value from neural activity across CANN layers using the population vector method, expressed as:

$$\theta^d = \frac{\int \theta\langle r(\theta,t)\rangle d\theta}{\int \langle r(\theta,t)\rangle d\theta} \quad (6)$$

where $\langle r(\theta,t)\rangle$ is the averaged firing rate of neurons at $\theta$ during the cueing period.

## B Experimental Paradigm and Statistical Methods

**Post-cueing paradigm.** Each trial consists of a first stimulus ($S_1$) lasting 200 ms, an inter-stimulus interval (ISI) lasting 1000 ms, a second stimulus ($S_2$) lasting 200 ms, a delay period of 3400 ms, a retrieval cue ($S_i, i = 1,2$) lasting 500 ms, and an inter-trial interval (ITI) of 1000 ms. In the single-layer CANN, $S_2$ was recalled during the cue phase to examine the influence of $S_1$ on $S_2$ within a trial. Simulations were performed for a single trial. In the two-layer CANN, the effects within and between trials were examined by recalling $S_1$ or $S_2$ during the cue phase in a randomized order across 100 consecutive trials. Simulations were conducted with 20 participants, each with various neuronal synaptic connection strengths as in Appendix A. For each participant, 100 trials were conducted.

**Stimulus orientation settings.** Orientations were selected from a uniform distribution over [-90°, 90°) with a step size of 1°. $\Delta S$ refers to the angular difference between the previous stimulus and the current stimulus. To ensure an equal probability of its occurrence, $\Delta S_{\text{within}}$ and $\Delta S_{\text{between}}$ were randomly generated with equal probability over [-90°, 90°]. Take the first trial in the two-layer CANN as an example, $\theta_{\text{cue}}^{s,1}$ was selected from a uniform distribution over [-90°, 90°) randomly, then $\theta_{\text{uncue}}^{s,1}$ and $\theta_{\text{cue}}^{s,2}$ were calculated:

$$\theta_{\text{uncue}}^{s,1} = \theta_{\text{cue}}^{s,1} + \Delta S_{\text{within}} \quad (7)$$

$$\theta_{\text{cue}}^{s,2} = \theta_{\text{cue}}^{s,1} - \Delta S_{\text{between}} \quad (8)$$

Subsequent stimuli were generated in the same way. For the $nth$ trial, $\theta_{\text{uncue}}^{s,n}$ of the current trial and $\theta_{\text{cue}}^{s,n+1}$ of the subsequent trial were calculated according to the following formulas:

$$\theta_{\text{uncue}}^{s,n} = \theta_{\text{cue}}^{s,n} + \Delta S_{\text{within}} \quad (9)$$

$$\theta_{\text{cue}}^{s,n+1} = \theta_{\text{cue}}^{s,n} - \Delta S_{\text{between}} \quad (10)$$

$\theta_{\text{cue}}^n$ and $\theta_{\text{uncue}}^n$ were assigned to $S_1$ or $S_2$ randomly, meanwhile, the cue index in $S_{\text{cue}}^n$ was determined.

**Behavioral readout.** In a single-layer CANN, external signals are input into the network, and the output is read from the same network using the population vector method. In a two-layer CANN, external signals are input into the lower layer, and the output is read from the higher layer.

Table 1: Model Parameters

| Parameter | Meaning | Value |
|---|---|---|
| **General Network Parameters** | | |
| $N$ | Number of neurons | 100 |
| $\tau$ | Time constant of synaptic current | 0.01 s |
| $\mu_J$ | Neuronal interaction noise strength | 0.01 |
| **Parameters for STD-dominated single-layer CANN (Fig. 2A-C)** | | |
| $J^0$ | Maximum synaptic connection strength | 0.13 |
| $a$ | Range of neuronal interaction | 0.5 |
| $k$ | Global inhibition strength | 0.0018 |
| $\mu_b$ | Background noise strength | 0.5 |
| $\tau_d$ | Time constant of $x$ | 3 s |
| $\tau_f$ | Time constant of $u$ | 0.3 s |
| $U_0$ | Increment of $u$ produced by a spike | 0.5 |
| **Parameters for STF-dominated single-layer CANN (Fig. 2D-F)** | | |
| $J^0$ | Maximum synaptic connection strength | 0.09 |
| $a$ | Range of neuronal interaction | 0.15 |
| $k$ | Global inhibition strength | 0.0095 |
| $\mu_b$ | Background noise strength | 0.5 |
| $\tau_d$ | Time constant of $x$ | 0.3 s |
| $\tau_f$ | Time constant of $u$ | 5 s |
| $U_0$ | Increment of $u$ produced by a spike | 0.2 |
| **Parameters for two-layer network (Fig. 3)** | | |
| $J^0_{LL}$ | Max synaptic connection strength (low-to-low) | 0.13 |
| $a_L$ | Range of neuronal interaction (low-level) | 0.5 |
| $k_L$ | Global inhibition strength (low-level) | 0.0018 |
| $\mu_L$ | Background noise strength (low-level) | 0.5 |
| $\tau_d^L$ | Time constant of $x$ (low-level) | 3 s |
| $\tau_f^L$ | Time constant of $u$ (low-level) | 0.3 s |
| $U_0^L$ | Increment of $u$ produced by a spike (low-level) | 0.5 |
| $J^0_{HL}$ | Max synaptic connection strength (low-to-high) | 0.02 |
| $a_{HL}$ | Range of neuronal interaction (low-to-high) | 0.15 |
| $J^0_{HH}$ | Max synaptic strength (high-to-high) | 0.09 |
| $a_H$ | Range of neuronal interaction (high-level) | 0.15 |
| $k_H$ | Global inhibition strength (high-level) | 0.0095 |
| $\mu_H$ | Background noise strength (high-level) | 0.5 |
| $\tau_d^H$ | Time constant of $x$ (high-level) | 0.3 s |
| $\tau_f^H$ | Time constant of $u$ (high-level) | 5 s |
| $U_0^H$ | Increment of $u$ produced by a spike (high-level) | 0.2 |
| **External Input** | | |
| $\alpha_{\text{sti}}$ | Strength of external stimulus | 20 |
| $a_{\text{sti}}$ | Spatial scale of external stimulus | 0.3 |
| $\mu_{\text{sti}}$ | Noise strength of external stimulus | 0.5 |
| $\alpha_{\text{cue}}$ | Strength of external cue | 2.5 |
| $a_{\text{cue}}$ | Spatial scale of external cue | 0.4 |
| $\mu_{\text{cue}}$ | Noise strength of external cue | 1 |

**Adjustment error.** For each trial, we measured the adjustment error as the angular distance between the cued stimulus angle and the decoded response angle.

$$\text{Error} = \theta^d_{\text{cue}} - \theta^s_{\text{cue}} \tag{11}$$

**Serial bias analysis.** Serial bias was calculated as the averaged adjustment error using a 30° sliding window, as a function of within-trial and between-trial angular differences. In Fig. 3D, E:

$$\Delta S_{\text{within}} = \theta_{\text{uncue}}^{s,n} - \theta_{\text{cue}}^{s,n} \tag{12}$$

$$\Delta S_{\text{between}} = \theta_{\text{cue}}^{s,n} - \theta_{\text{cue}}^{s,n+1} \tag{13}$$

In Fig. 3F, we further analyzed the average error as a function of the memory representation angular difference:

$$\Delta R_{\text{between}} = \theta_{\text{cue}}^{d,n} - \theta_{\text{cue}}^{s,n+1} \tag{14}$$

**DoG curve.** To measure the amplitude of serial dependence, we fit the error plot with the first derivative of a Gaussian curve (DoG) $G(x) = \sqrt{e}(x/\sigma)A_{\text{DoG}}\exp(-x^2/2\sigma^2)$, where $A_{\text{DoG}}$ is the amplitude of the curve peak, $\sigma$ is the curve width.

**Partial no-report paradigm.** To test whether serial dependence occurs without a prior reported stimulus, the cue was absent in 40% of the trials. Trials following those without a report were used to analyze serial biases.

**Computer resources.** We used Python 3.11.9 and Brainpy 2.6.0.post20240918 for simulations. All experiments were conducted on a consumer-grade desktop computer (AMD Ryzen 9 9950X, 32GB DDR5 RAM) with hour-scale runtimes.

# C Numerical Simulation of $u$ and $x$

**The fitting of $x$ curve in an STD-dominated single-layer CANN.** In an STD-dominated single-layer CANN, $x$ is fitted by the following equation:

$$x(\theta,t) = 1 - A_x^1(t)\exp\left[-(\theta-\theta_1^s)^2/2a^2\right] - A_x^2(t)\exp\left[-(\theta-\theta_2^s)^2/2a^2\right] \tag{15}$$

For example, in Fig. 2B, we set $\theta_1^s = -30°$, $\theta_2^s = 0°$ as the initial guess value. The amplitudes $A_x^1(t)$ and $A_x^2(t)$ decayed from the early delay period (Fig. S1A) to the late delay period (Fig. S1B). We calculated these parameters from the time point after $S_2$ disappeared ($t_0 = 1.5$ s) to the time point before the cue period ($t_e = 4.8$ s), shown as dots in Fig. S1C, and found that they decayed exponentially with a time constant $\tau_d$ (curve in Fig. S1C), described as:

$$A_x^i(t) = A_x^i(t_0)\exp\left[-(t-t_0)/\tau_d\right], i = 1, 2 \tag{16}$$

where $A_x^1(t_0) = 0.46, A_x^2(t_0) = 0.68$.

The above analysis is applicable to other combinations of $\theta_1^s$ and $\theta_2^s$.

Table 2: The fitting parameters of $x$ curve

| t (s) | $\theta_1^s$ (°) | $\theta_2^s$ (°) | $A_x^1$ | $A_x^2$ | $a$ |
|---|---|---|---|---|---|
| 2.0 | -32.72 | 6.13 | 0.39 | 0.57 | 0.31 |
| 4.5 | -32.72 | 6.13 | 0.17 | 0.25 | 0.31 |

**The fitting of $u$ curve in an STF-dominated single-layer CANN.** In an STF-dominated single-layer CANN, the $u$ curve was fitted by the following equation:

$$u(\theta,t) = A_u^1(t)\exp\left[-(\theta-\theta_1^s)^2/2a^2\right] + A_u^2(t)\exp\left[-(\theta-\theta_2^s)^2/2a^2\right] \tag{17}$$

In Fig. 2E, we fixed $\theta_1^s = -30°$ and $\theta_2^s = 0°$. The amplitude parameters $A_u^1(t)$ and $A_u^2(t)$ decayed from the early delay period (Fig. S1D) to the late delay period (Fig. S1E). We calculated these parameters from the time point after $S_2$ disappeared ($t_0 = 1.5$ s) to the time point before the cue period ($t_e = 4.8$ s), shown as dots in Fig. S1F, and found that they decayed exponentially with the time constant $\tau_f$ (curve in Fig. S1F), described as:

$$A_u^i(t) = A_u^i(t_0)\exp\left[-(t-t_0)/\tau_f\right], i = 1, 2 \tag{18}$$

where $A_u^1(t_0) = 0.09, A_u^2(t_0) = 0.12$.

The above analysis is applicable to other combinations of $\theta_1^s$ and $\theta_2^s$.

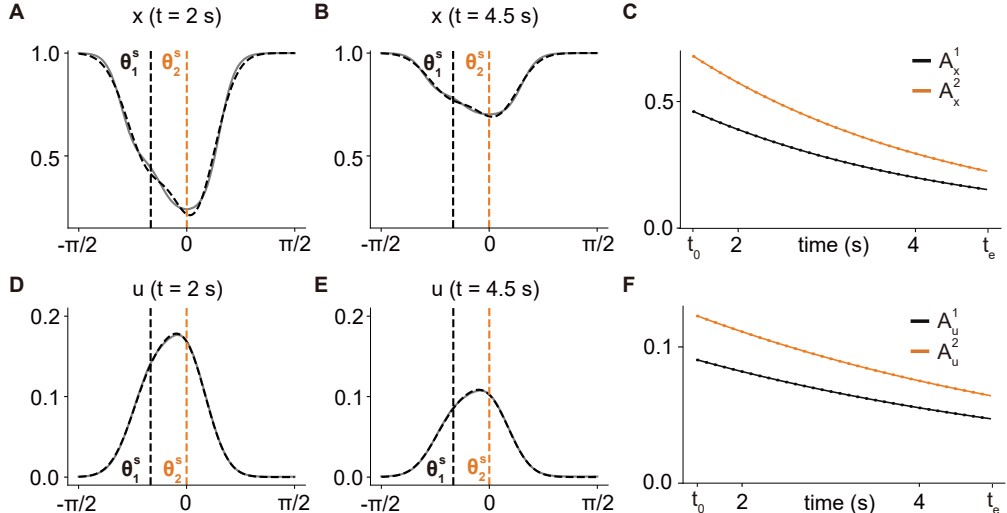

Figure S1: The fitting of $x$ and $u$ curves in a single-layer CANN. (A-C) $x$ in an STD-dominated single-layer CANN. (D-F) $u$ in an STF-dominated single-layer CANN. (A, D) Simulation (gray curve) and fitting results (dotted curve) in the early delay period (t = 2.0 s). (B, E) Simulation (gray curve) and fitting results (dotted curve) the late delay period (t = 4.5 s). (C, F) Exponential decay during the delay period. Dots represent the simulation results, and curves represent the exponential fitting results.

Table 3: The fitting parameters of $u$ curve

| t (s) | $\theta_1^s$ (°) | $\theta_2^s$ (°) | $A_u^1$ | $A_u^2$ | $a$ |
|-------|-----------|-----------|---------|---------|------|
| 2.0 | -30.00 | 0.00 | 0.08 | 0.11 | 0.31 |
| 4.5 | -30.00 | 0.00 | 0.05 | 0.07 | 0.31 |

**The fitting of $x_L$ and $u_H$ curve in a two-layer CANN.** In a two-layer CANN, the $x_L$ curve in the $(n+1)th$ trial was fitted by the following equation:

$$x_L(\theta_L, t) = 1 - A_x^1(t) \exp\left[-(\theta_L - \theta_1^{s,n+1})^2/2a_x^2\right] - A_x^2(t) \exp\left[-(\theta_L - \theta_2^{s,n+1})^2/2a_x^2\right]$$
$$- A_x^{\text{cue}}(t) \exp\left[-(\theta_L - \theta_{\text{cue}}^{s,n})^2/2a_x^2\right] \tag{19}$$

In Fig. 4B, we set $\theta_{\text{cue}}^n = 20°$, $\theta_1^{n+1} = -30°$, and $\theta_2^{n+1} = 0°$ as the initial guess values. The amplitude parameters $A_x^{\text{cue}}(t)$, $A_x^1(t)$ and $A_x^2(t)$ decayed from the early delay period (Fig. S2A) to the late delay period (Fig. S2B). We calculated the fitted parameters from the time point after $S_2^{n+1}$ disappeared ($t_0 = 7.8$ s) to the time point before the cue period ($t_e = 11.1$ s), shown as dots in Fig. S2C, and found that they decayed exponentially with a time constant of $\tau_d$ (curve in Fig. S2C), which can also be described as:

$$A_x^i(t) = A_x^i(t_0) \exp\left[-(t - t_0)/\tau_d\right], i = 1, 2, \text{cue} \tag{20}$$

where $A_x^{\text{cue}}(t_0) = 0.23, A_x^1(t_0) = 0.44, A_x^2(t_0) = 0.68$.

The above analysis is applicable to other combinations of $\theta_{\text{cue}}^n$, $\theta_1^{n+1}$ and $\theta_2^{n+1}$.

Table 4: The fitting parameters of $x_L$ curve

| t (s) | $\theta_{\text{cue}}^{d,n}$ (°) | $\theta_1^{s,n+1}$ (°) | $\theta_2^{s,n+1}$ (°) | $A_x^{\text{cue}}$ | $A_x^1$ | $A_x^2$ | $a_x$ |
|-------|-----------|-----------|-----------|------|------|------|------|
| 8.3 | 39.13 | -35.41 | 3.26 | 0.19 | 0.37 | 0.57 | 0.33 |
| 10.8 | 39.13 | -35.41 | 3.26 | 0.08 | 0.16 | 0.25 | 0.33 |

$u_H$ curve in the $(n + 1)th$ trial can be fit by the following equation:

$$u_H(\theta_H, t) = A_u^1(t) \exp\left[-(\theta_H - \theta_1^{s,n+1})^2/2a_u^2\right] + A_u^2(t) \exp\left[-(\theta_H - \theta_2^{s,n+1})^2/2a_u^2\right]$$
$$+ A_u^{\text{cue}}(t) \exp\left[-(\theta_H - \theta_{\text{cue},H}^{d,n})^2/2a_u^2\right] \tag{21}$$

In Fig. 3C and Fig. 4B, we fixed $\theta_{\text{cue}}^n = 20°$, $\theta_1^{n+1} = -30°$, and $\theta_2^{n+1} = 0°$. $A_u^{\text{cue}}(t)$, $A_u^1(t)$ and $A_u^2(t)$ decayed from the early delay period (Fig. S2D) to the late delay period (Fig. S2E). We calculated these parameters from the time point after $S_2^{n+1}$ disappeared ($t_0 = 7.8$ s) to the time point before the cue period ($t_e = 11.1$ s), shown as dots in Fig. S2F, and found that they decayed exponentially with the time constant $\tau_f$ (curve in Fig. S2F), which can also be described as:

$$A_u^i(t) = A_u^i(t_0) \exp\left[-(t - t_0)/\tau_f\right], i = 1, 2, \text{cue} \tag{22}$$

where $A_u^{\text{cue}}(t_0) = 0.22$, $A_u^1(t_0) = 0.14$, $A_u^2(t_0) = 0.08$.

The above analysis is applicable to other combinations of $\theta_{\text{cue}}^n$, $\theta_1^{n+1}$ and $\theta_2^{n+1}$.

Table 5: The fitting parameters of $u_H$ curve

| t (s) | $\theta_{\text{cue}}^{d,n}$ (°) | $\theta_1^{s,n+1}$ (°) | $\theta_2^{s,n+1}$ (°) | $A_u^{\text{cue}}$ | $A_u^1$ | $A_u^2$ | $a_u$ |
|---|---|---|---|---|---|---|---|
| 8.3 | 20.00 | -30.00 | 0.00 | 0.20 | 0.12 | 0.07 | 0.31 |
| 10.8 | 20.00 | -30.00 | 0.00 | 0.31 | 0.08 | 0.04 | 0.31 |

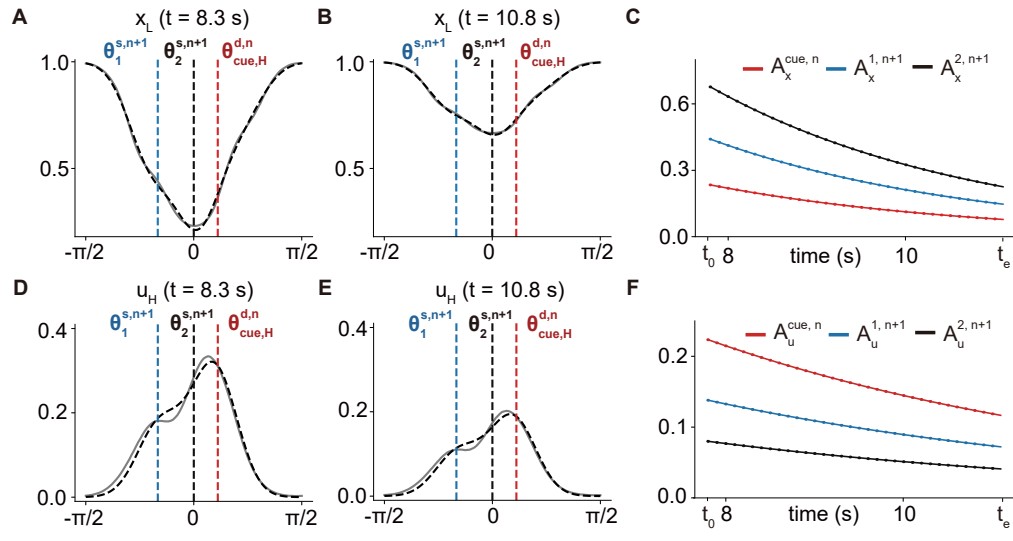

Figure S2: The fitting of $x_L$ and $u_H$ curves in a two-layer CANN. (A-C) $x_L$ in an STD-dominated low layer. (D-F) $u_H$ in an STF-dominated high layer.

## D Dynamics of STP Variables

In Fig. 2A, we illustrated only the dynamics of the available neurotransmitter $x(\theta, t)$ to emphasize that, under the STD-dominated condition ($\tau_d \gg \tau_f$), the slow recovery of $x(\theta, t)$ is the main factor producing the repulsive effect. Here, the release probability $u(\theta, t)$ rapidly decays to zero after stimulus presentation and contributes minimally (Fig. S3A, top). Upon presentation of the retrieval signal, the instantaneous synaptic efficacy $u(\theta, t)x(\theta, t)$ shows an asymmetric distribution—the side near the previous stimulus $S_1$ is weaker (Fig. S3A, bottom), resulting in repulsion.

In Fig. 2D, we illustrated only the dynamics of the release probability $u(\theta, t)$ to highlight facilitation accumulation under the STF-dominated condition ($\tau_f \gg \tau_d$). Here, $x(\theta, t)$ rapidly recovers to baseline and contributes minimally (Fig. S3B, top). Upon retrieval, the synaptic efficacy $u(\theta, t)x(\theta, t)$ is enhanced near $S_1$ (Fig. S3B, bottom), producing an attractive bias.

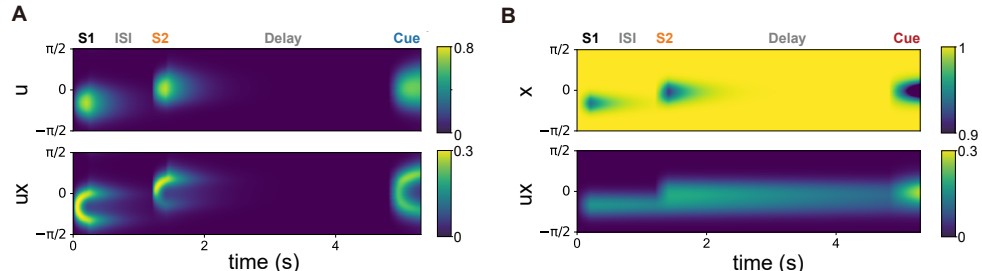

Figure S3: Dynamics of STP components in one-layer networks. (A) STD-dominated condition ($\tau_d \gg \tau_f$). Top: release probability $u(\theta, t)$ rapidly decays. Bottom: synaptic efficacy $u(\theta, t)x(\theta, t)$ is weaker near $S_1$. (B) STF-dominated condition ($\tau_f \gg \tau_d$). Top: available neurotransmitter $x(\theta, t)$ rapidly recovers. Bottom: synaptic efficacy $u(\theta, t)x(\theta, t)$ is enhanced near $S_1$.

## E Control Analysis on the Recalled Stimulus in the Single-layer CANN

In the one-layer model (Fig. 2), only the second stimulus ($S_2$) was cued to examine the influence of the preceding one ($S_1$) on the subsequent one ($S_2$) based on the definition of *serial dependence*. To confirm that this choice does not qualitatively affect the model behavior, we conducted additional control experiments in which the one-layer model was re-run with random cueing of either $S_1$ or $S_2$ in interleaved trials.

The results showed that under STD-dominance, the model consistently exhibited a significant repulsive bias ($A_{\mathrm{DoG}} = -3.51°$; $t(19) = 168.57$, $p < .001$, Fig. S4A), identical in sign and statistical significance to the repulsion reported in Fig. 2C. Under STF-dominance, the model consistently exhibited a significant attractive bias ($A_{\mathrm{DoG}} = 1.68°$; $t(19) = 96.97$, $p < .001$, Fig. S4B), identical to the attraction reported in Fig. 2F. These findings demonstrate that the choice of the recalled stimulus ($S_1$ or $S_2$) does not alter the qualitative pattern of serial dependence, confirming the robustness of our main conclusions.

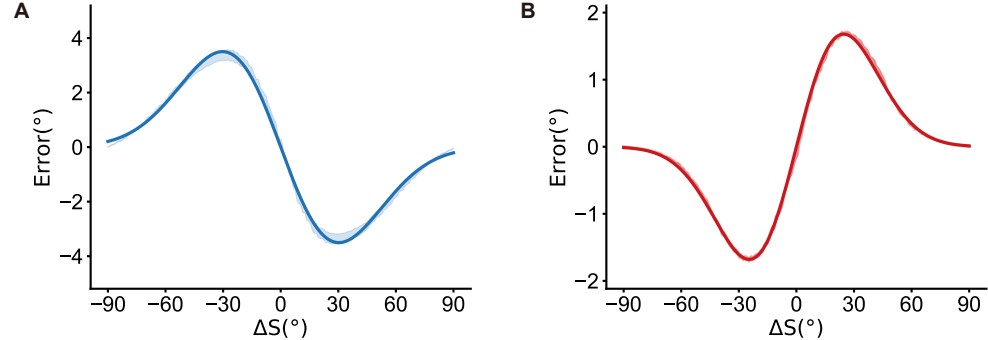

Figure S4: Control analysis of the recalled stimulus in the one-layer model. (A) STD-dominated single-layer CANN re-run with randomly cued S1 or S2 in interleaved trials. The adjustment error curve showed a significant repulsive bias. (B) STF-dominated single-layer CANN re-run with randomly cued S1 or S2 in interleaved trials. The adjustment error curve showed a significant attractive bias.

## F Readout Rules

In our main analyses, we decoded orientation from neural activity via population vector method (PVM, Appendix A). To evaluate potential readout bias from asymmetrical responses, we reanalyzed Fig. 3D using three other methods:

**Center-of-Mass (COM).** The center-of-mass (COM) decoding computes the firing-rate-weighted circular mean of the population activity, implemented explicitly as a center-of-mass estimator on the

unit circle:

$$\hat{\theta}_{\text{COM}} = \arg\left(\sum_i r_i e^{j\theta_i}\right).$$

In our simulations, the COM and population vector method (PVM) yield identical estimates. This equivalence arises because the tuning curves are symmetric, densely and uniformly distributed across the stimulus space, and without truncation or nonlinearity. Under such conditions, both COM and PVM effectively perform a linear weighted average of the population activity.

**Maximum Likelihood (ML).** We assume that each neuron's firing rate follows a Poisson distribution with a mean $\lambda_i(\theta)$ given by its tuning curve. The most likely stimulus $\hat{\theta}_{\text{ML}}$ maximizes the log-likelihood function:

$$\hat{\theta}_{\text{ML}} = \arg\max_\theta \sum_i \left[r_i \log \lambda_i(\theta) - \lambda_i(\theta) - \log(r_i!)\right].$$

We evaluated the log-likelihood across all candidate $\theta$ values and selected the one that yielded the maximum.

**Peak Decoding (Peak).** As a minimal heuristic, we also tested a peak decoder, which simply selects the preferred stimulus of the neuron with the highest firing rate:

$$\hat{\theta}_{\text{Peak}} = \theta_{i^*}, \quad i^* = \arg\max_i r_i.$$

All three methods produced consistent results with PVM, confirming robust within-trial repulsion (serial bias amplitudes/curve peak at: PVM: -0.90°/37.92°, Fig. 3D; COM: -0.90°/37.92°, Fig. S5A; ML: -0.97°/36.97°, Fig. S5B; Peak: -0.61°/45.34°, Fig. S5C) and between-trial attraction (PVM: 1.17°/19.34°; COM: 1.17°/19.34°; ML: 1.05°/19.86°; Peak: 1.93°/18.21°). These results demonstrate that the qualitative pattern of serial dependence is robust to the choice of readout rule.

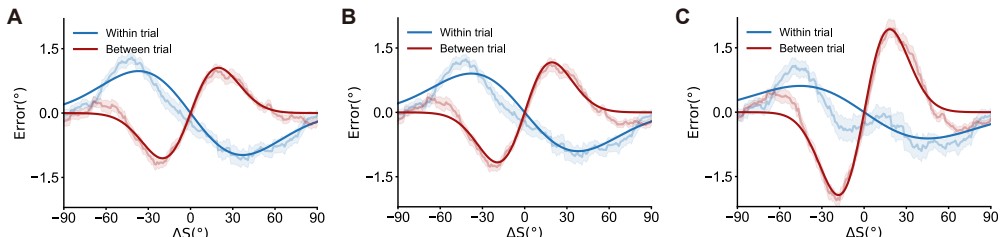

Figure S5: Comparison of different readout rules for decoding perceptual decisions. (A) Center-of-mass (COM) decoding. (B) Maximum likelihood (ML) decoding. (C) Peak decoding.

## G Importance of the Order Between STD and STF Layers

To examine the importance of the order of short-term depression (STD) and short-term facilitation (STF) in the model architecture, we conducted control simulations with reversed synaptic configurations. A network with a lower STF-dominated layer ($\tau_f = 5$, $\tau_d = 0.3$) and a higher STD-dominated layer ($\tau_d = 3$, $\tau_f = 0.3$) exhibited exclusively attractive serial biases. Both within-trial effects and between-trial effects showed robust attraction (Fig. S6), confirming that the functional synergy between sensitivity and stability critically depends on the correct sequential order of STD and STF layers.

## H Parameter Sensitivity Analysis

To evaluate the robustness of the model with respect to the choice of short-term plasticity (STP) parameters, we conducted a series of sensitivity analyses. The time constants for STD-dominated and STF-dominated synapses $\tau$s were selected based on previous physiological studies[47, 48, 53, 50] . As these parameters are critical for shaping the network dynamics, we varied their absolute values while preserving their dominance relationship (i.e., $\tau_d \gg \tau_f$ for STD, and $\tau_f \gg \tau_d$ for STF).

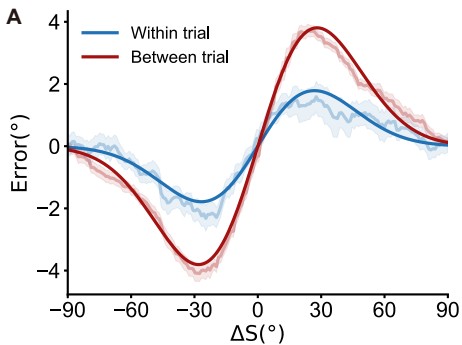

Figure S6: Effect of reversing STD/STF order on serial dependence bias. Both within-trial and between-trial bias curves showed exclusively attractive effects.

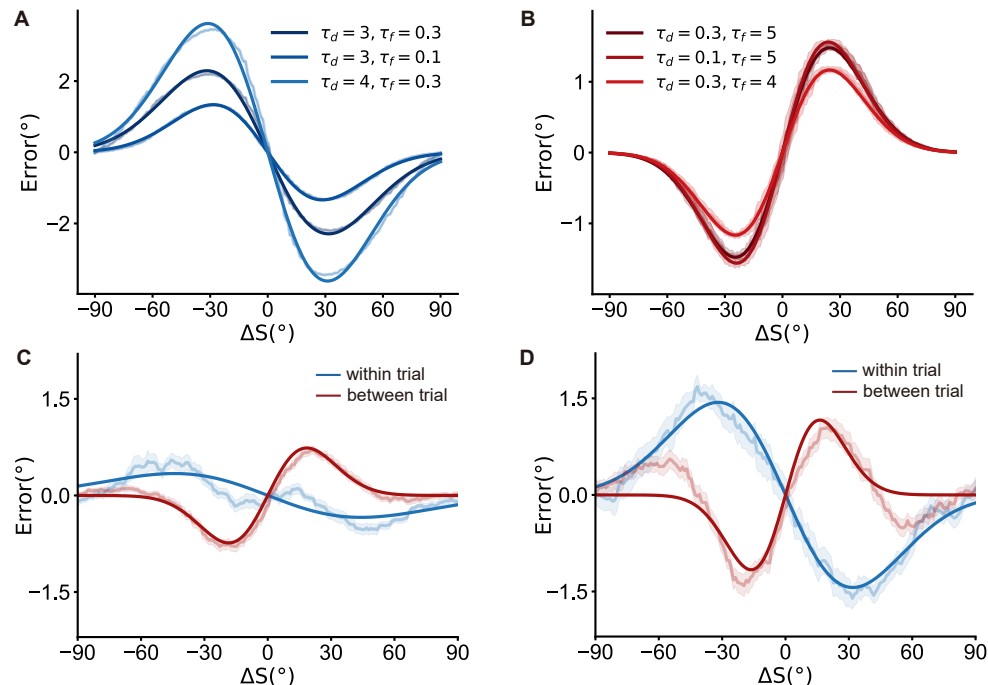

Figure S7: Sensitivity of serial dependence effects to short-term plasticity (STP) parameters. (A) Robustness of STD-dominated synapses in producing repulsive bias. All parameter sets showed significant repulsion effects. (B) Robustness of STF-dominated synapses in producing attractive bias. All parameter sets showed significant attraction effects. (C–D) Robustness of serial dependence in two-layer networks. In both cases, within-trial repulsion and between-trial attraction remained significant, differing only in effect magnitude.

**Robustness of STD dominance in inducing repulsion effects.** For the STD-dominated condition (Fig. 2C), we maintained $\tau_d \gg \tau_f$ and compared two sets of parameters ($\tau_f = 0.1$, $\tau_d = 3$ vs. $\tau_f = 0.3$, $\tau_d = 4$). In both cases, the model exhibited a statistically significant repulsion effect (both $t(19) > 213.97$, $p < .001$), with only the magnitude of repulsion varying ($-2.29°$ in Fig. 2C, $-1.34°$ and $-3.61°$ in Fig. S7A).

**Robustness of STF dominance in inducing attraction effects.** For the STF-dominated condition (Fig. 2F), we maintained $\tau_f \gg \tau_d$ and compared two sets of parameters ($\tau_f = 5$, $\tau_d = 0.1$ vs. $\tau_f = 4$, $\tau_d = 0.3$). In both cases, the model exhibited a statistically significant attraction effect (both $t(19) > 85.12$, $p < .001$), with only the magnitude of attraction varying ($1.48°$ in Fig. 2F, $1.56°$ and $1.16°$ in Fig. S7B).

**Robustness of parameters in two-layer networks.** Using the same ISI and ITI parameters as in Fig. 3D, we adjusted the absolute values of $\tau_d$ and $\tau_f$ in the low and high layers while preserving their relative relationships ($\tau_d^L \gg \tau_f^L$ and $\tau_f^H \gg \tau_d^H$). Serial dependence effects were computed for two parameter sets ($\tau_d^L = 2$, $\tau_f^H = 4$ in Fig. S7C; $\tau_d^L = 4$, $\tau_f^H = 6$ in Fig. S7D). The results showed that within-trial repulsion and between-trial attraction effects remained statistically significant (within-trial: both $t(19) > 5.52$, $p < .001$; between-trial: both $t(19) > 3.48$, $p < .01$), with only their magnitudes varying (within-trial: $-0.34°$ and $-1.44°$; between-trial: $0.73°$ and $1.16°$).

Our results demonstrated that, although changes in absolute values of $\tau_d$ and $\tau_f$ modulate the strength of sequence-dependent effects, it is their relative relationship (i.e., whether STD or STF dominance) that determines the qualitative nature of repulsion and attraction.

## I Effect of Synaptic Heterogeneity on Model Performance

To explore how the heterogeneity of STP affects model performance, we re-ran the model with an inter-stimulus interval (ISI) of 1 s and applied $\pm 10\%$ random perturbations to STP parameters ($\tau_d$ and $\tau_f$) of all synapses. Under STD-dominance, the adjustment error curve (variant STD, Fig. S8A) aligned with that in Fig. 2C (uniform STD, Fig. S8A). Results showed minimal amplitude change (from $-2.29°$ to $-2.28°$) and minimal DoG peak shift (from $31.71°$ to $31.78°$). Under STF-dominance, the adjustment error curve (variant STF, Fig. S8B) also closely matched that in Fig. 2F (uniform STF, Fig. S8B). Results showed stable attraction amplitude ($1.48°$) and minimal DoG peak shift (from $24.49°$ to $24.41°$). These results indicate that moderate heterogeneity in STP has a negligible impact on the qualitative pattern of serial dependence.

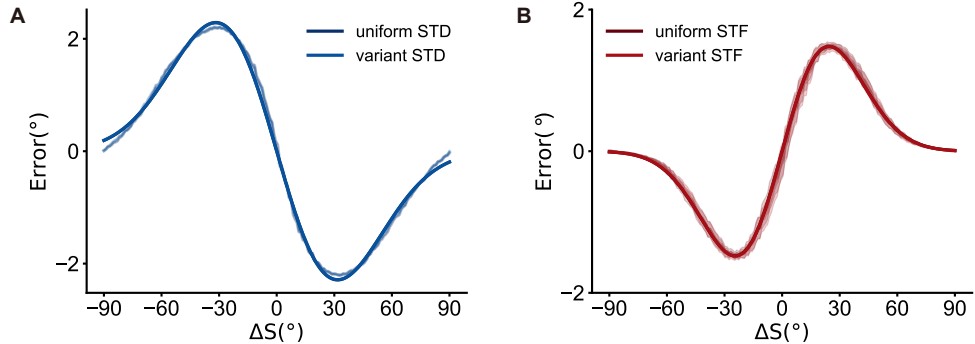

Figure S8: Effect of synaptic heterogeneity on model performance. (A) STD-dominated single-layer CANN re-run with uniform or variant $\tau_d$. (B) STF-dominated single-layer CANN re-run with uniform or variant $\tau_f$.

