# OpenReview forum: "Neural Correlates of Serial Dependence: Synaptic Short-term Plasticity Orchestrates Repulsion and Attraction"
_NeurIPS.cc/2025/Conference — NeurIPS 2025 poster_

### Official Review · Reviewer_rAyR · 2025-06-17

**Clarity:** 4
**Significance:** 3
**Originality:** 3
**Rating:** 5
**Confidence:** 3

**Summary:**

In this study, the authors present a two-stage continuous attractor network model endowed with short term facilitation and depression, capable of explaining psychophysical results regarding the sequential dependence upon previous stimuli of perceptual judgments in task participants.

**Questions:**

Major concerns:

1.	If the present model were really compatible with Bayesian accounts of sensory representation, shouldn’t its properties be in some way dependent on the statistics of the stimuli presented to the network? As it is, the short-term plasticity timescales and parameters are fixed by the modeler, whereas a serially dependent Bayesian prior over network activity states should theoretically be related to the timescale statistics of presented stimuli. Is there any room for dynamic adaptation of these parameters within the model, or any experimental evidence in favor of such a thing? As it stands, the present model seems to me to be a mechanistic alternative account for empirical data that is likely not compatible with Bayesian approaches; there ought to be experiments that could be used to disambiguate these two classes of model (or model modifications that could produce more synchrony), and a discussion of these points could be very beneficial for the community.

2.	Relatedly, does the present model make any testable predictions that could be used to validate it? Figure 5 seems to provide some traction on this question, but a more detailed discussion would be beneficial. In particular, how dependent are these results on the (as far as I can tell) arbitrarily selected timescales of STF and STD? More importantly, if this model were really underlying the psychophysical results observed, shouldn’t there be a near-exact relationship between the timescales of STF and STD in early sensory areas and PFC and the timescales of within-trial repulsion/ across-trial attraction?

Minor concerns:

Line 151: the \Delta S change reported must be highly parameter-dependent, correct? The parameter-sensitivity of this result should be discussed (as well as its implications for testable model predictions as mentioned above).

Line 226: citations need for the gain and two-process models.

**Ethical Concerns:**

["NO or VERY MINOR ethics concerns only"]

**Final Justification:**

The authors did a good job addressing my concerns and proposing concrete modifications to their manuscript that will accordingly improve its quality. My initial assessment of the paper was 'Accept', and this remains my assessment.

**Limitations:**

Yes

**Quality:**

3

**Strengths And Weaknesses:**

The paper is well-written and clear, with an appropriate amount of time spent constructing the two-stage model and analyzing its properties. The results are interpretable, intuitive, and thorough; the model is also reasonably well substantiated by a literature review and appears (as far as I can tell) original. I have some concerns about the authors’ attempt to relate their model to a Bayesian interpretation of serial dependence, explained below. While I do think that the model is quite well designed to explain the specific set of experimental results that the authors have selected, I have a hard time seeing how a model of this kind could be adapted to explain or provide predictions for neural responses or human reports in response to more complex, naturalistic stimuli, i.e. I view it as fundamentally a ‘toy’ model, that leaves little clue for how it could be extended to more complex situations.

---

> ### Author Rebuttal · Authors · 2025-07-30
>
> Thank you for the valuable and constructive comments, and the recognition of the clarity, interpretability, and originality of our work. In the below, we would like to clarify your concerns one-by-one.
>
> **Weakness:**
>
> 1.	About the generality of the model
>
> As you noted, our model targets on explaining the serial dependence phenomena on simple physical features like visual orientation, which is a common strategy in the cognitive neuroscience field of using low-dimensional, well-controlled stimuli to isolate the core neural mechanisms. The simple features used in psychophysical experiments motivated us to build relatively simple network models. Nevertheless, we expect that this simple model gives us insight into the fundamental mechanism of serial dependence, that is, short-term synaptic plasticity, through distinct time constants for depression and facilitation, generating varying serial dependence biases.
>
> Recently, serial dependence experiments have been extended to include complex stimuli such as facial emotion, identity, and body size (e.g., Manassi et al., Journal of Vision, 2023), and similar bias phenomena have been observed. Considering that STP is a general property of neural systems and that STP can explain serial dependence with simple features, we strongly believe that the STP-based mechanism proposed in this work is also applicable to these complex features. However, to validate this, we need to first build much more complicated network models mimicking how the brain processes these complex features. Our present work can be regarded as the first step in demonstrating the feasibility of the STP-based mechanism for serial dependence, and in future work, we will extend the model to more complex situations.
>
> **Questions:**
>
> Major concerns:
>
> 1.	About the link to Bayesian interpretation
>
> Thanks for raising this important question. Please note that in serial dependence research, when people claim Bayesian interpretation, they do not refer to how sensory representation depends on the stimulus statistics over long time, but rather emphasize on how perception depends on the recent history of stimulus sequence, which falls into the time scale of STP. Specifically, the Bayesian theory of serial dependence consists of two stages of processing (Pascucci et al., Plos Biology, 2019).
> (1)	Sensory Encoding Stage: The Bayesian theory posits that, after exposure to the first stimulus ($\theta_n$, Fig. 4A bottom), neurons preferring stimuli near $\theta_n$ undergo adaptation, manifested as reduced responses to the second stimulus. This alters the likelihood function in the Bayesian framework and generates the repulsion effect in sensory perception.
> In the first layer of our model, due to STD-dominance, the available neurotransmitter $x$ decays after the presence of the first stimulus, and the distribution of $x$ values across the neuron population is non-uniform (black curve in Fig. 4B bottom), with more active neurons having smaller values of $x$. When the second stimulus is presented, this non-uniform distribution of $x$ across the neuron population induces alternated responses compared to that to the first stimulus, which can be regarded as the likelihood function being changed in the Bayesian framework.
>
> (2)	Post-Perception Decision Stage: The Bayesian theory posits that higher cortical areas develop weight bias toward the prior stimulus ($\theta_n$, Fig. 4A top), modifying the prior distribution and inducing the attraction effect.
> In the second layer of our model, due to STF, the neurotransmitter release probability $u$ (green curve, Fig. 4B middle) increases at the location of the previous stimulus ($\theta_{cue}^{d,n}$), which facilitates neuronal responses at the previous stimulus location, as if the prior distribution of stimulus is changed in the Bayesian framework.
> In summary, unlike the conventional Bayesian models, which posit priors are shaped by long-term environmental statistics (e.g., Wei et al., Nature Neuroscience, 2015), the Bayesian theory of serial dependence considers experience-dependent modulations of likelihood function and prior in short timescale, and our model provides a potential neural mechanism to support this view. Since the Bayesian view of serial dependence has been recognized in the field, we hope that linking our model to this view will strengthen our understanding of the neural mechanism of serial dependence. We will elaborate on this connection in the revised Discussion section.
>
> 2.	About model prediction
>
> Yes, our model generates testable predictions and they can be validated. Specifically, our model proposes that $\tau_d$ in sensory areas governs the within-trial repulsion-to-attraction transition, while $\tau_f$ in high layers constrains the magnitude of across-trial attraction. These predictions address the reviewer's central query regarding hierarchical timescale dependencies. Please note that in our model, it is the relative relationship between $\tau_d$ and $\tau_f$, rather than their absolute values, that determines the serial dependence biases. The neuroscience data has shown that STF dominates in PFC while STD dominates in sensory cortex (STD: Abbot et al., Science, 1997; Chung et al., Neuron, 2002; Yu et al., Neuron, 2022; STF: Markram et al., Nature, 1996; Wang et al., Nature Neuroscience, 2006).
>
> As concerned by the reviewer, we conducted extra experiments to analyze the model parameter dependence. The results show that the emergence of repulsive effect primarily depends on the relative relationship $\tau_d \gg \tau_f$ in the lower-layer of the model, whereas the attractive effect relies on the $\tau_f \gg \tau_d$ relationship in the upper-layer. The absolute values of $\tau_d$ and $\tau_f$ are not crucial, and they only modulate the amplitude of the bias effect when ISI/ITI are fixed. We will add discussions about this point in the revised manuscript.
>
> Minor concerns:
>
> 1. Thanks for the question. $\Delta S$ denotes the difference between the prior and current stimuli. In the classical serial dependence curves, both attractive and repulsive biases exhibit a peak around $\Delta S$ of approximately 30°. In other words, the bias is strongest when the historical sensory information is about 20°–40° away from the current stimulus.
> Regarding the value of $\Delta S$ generating the strongest bias, our study showed that it mildly depends on STP parameters. For instance, in the STD-dominance condition (Fig. 2C), when we varied the absolute values of $\tau_d$ and $\tau_f$ while keeping $\tau_d \gg \tau_f$ holds ($\tau_f$=0.1, $\tau_d$=3 vs. $\tau_f$=0.3, $\tau_d$=2), we observed only minor shift in the peak repulsion angle (from 31.6° in Fig.2C to 28.5° and 32.9°, respectively). We also studied this sensitivity on STP time constants in both single- and two-layer networks, finding that the STP parameter variation does not alter the characteristic peak $\Delta S$ value. We will report these results in the revised manuscript.
>
> 2. Thanks for the suggestion. We will add citations for the gain model (Fischer et al., Nature Neuroscience, 2014) and two-process model (Schwiedrzik, Melloni et al., Cerebral Cortex, 2014; Pascucci et al., PLOS Biology, 2019) in the revised manuscript.
>
> Finally, we thank the reviewer for carefully reviewing our manuscript and providing many constructive suggestions, which are highly valuable to improve our work, and we commit to incorporating the corrections and new analyses into the final version of the paper. We hope our replies have addressed all concerns of the reviewer, and if not, please feel free to discuss with us.

---

> > ### Comment · Reviewer_rAyR · 2025-08-04
> >
> > Thank you, these comments address my concerns. I will maintain my score, which was already 'Accept'.

---

> > > ### Author Response · Authors · 2025-08-05
> > >
> > > Thanks for the support of our work!

---

### Official Review · Reviewer_fkF9 · 2025-07-01

**Clarity:** 3
**Significance:** 3
**Originality:** 3
**Rating:** 4
**Confidence:** 4

**Summary:**

In psychophysics and behavioral science, there is by now a large literature documenting the biases in perceptual decision related to the stimulus history. This paper proposed a neural network model to explain these serial dependence effects. Their main model is a two-layer coupled continuous attractor network (CANN), with plasticity in the connectivity. The first layer of the network was assumed to exhibit synaptic depression (STD), while the second layer was assumed to exhibit synaptic facilitation (STF). The network was shown to generate both attractive biases and repulsive biases. Both types of biases have been reported in the previous experimental studies.

**Questions:**

How was perceptual decision modeled in the network? Different readout rules may lead to different results as the population activity profile may be asymmetric. It would be interesting to examine this question.

Would it be possible to show some direct comparisons to example experimental results?

Are there any patterns of biases reported in experiments that the current model can not explain? If the answer is yes, it would be useful to discuss these as well.


Some more questions about the choice of the models:

(i) Would it be possible for a single-layer network to explain both attraction/repulsion?

(ii) Is the order the the STD and STF important? That is, would a model with STF in the first layer and STD in the second layer lead to different predictions than those shown in the paper?

(iii) Is the attractor dynamics important in the model? If not, one may be able to derive a more general network mechanism.

**Ethical Concerns:**

["NO or VERY MINOR ethics concerns only"]

**Final Justification:**

The rebutal addressed most of the concerns I had. The authors proposed several revisions that would make the paper stronger.  I increased the score for “quality” from 3 to 4.

I will keep my overall rating, which is already an “accept”. I think this paper should be a nice contribution for NeurIPS.

**Limitations:**

The  limitations of the work were properly discussed.

**Paper Formatting Concerns:**

None.

**Quality:**

4

**Strengths And Weaknesses:**

Strengths:

Overall, this is a solid paper that makes an interesting contribution.

- The paper is well written. Although the work is technical, it is relatively easy to follow. The logic of the paper is generally clear.

- The literature on the serial biases in psychophysics is quite messy. Given that, it is particularly nice to see models attempting to unify these experimental findings into a unifying computational framework.

- The proposed network model can simultaneously explain both attractive biases and repulsive biases.

- The model generates direct predictions that can be tested in future experiments (Fig. 5).



Weaknesses:

- The dependence of the results on the choice of key model parameters (e.g., the time constants of STD/STF) was not discussed. Thus, the robustness of these findings remain unclear.

- While qualitatively comparisons to the experimental results were provided in the paper, there was no direct comparison to the data.

- Line 82-85 motivates the model by citing papers showing STD in sensory areas and LTP in high-level areas. I believe these experimental data are only suggestive. It remains largely unclear what happens regarding STD/STF when the brain is performing some tasks. Can the authors tone this statement down? [minor weakness]

---

> ### Author Rebuttal · Authors · 2025-07-31
>
> Thank you for the positive, encouraging and valuable comments, and constructive questions. In the below, we would like to clarify your concerns one by one.
>
> **Weakness:**
>
> 1.	About the dependence of the results on the choice of key model parameters:
> Thank you for raising this important issue.
> Our choice of time constants $\tau$s for STD-dominance and STF-dominance is based on the biological data (see e.g., Abbot et al., Science, 1997; Chung et al., Neuron, 2002; Yu et al., Neuron, 2022; Wang et al., Nature Neuroscience, 2006).
> But as pointed out by the reviewer, since these parameters are so critical, it is worth analyzing the robustness of our findings on their values. We therefore conducted the robustness analyses. For example, for the STD-dominance condition (Fig.2C), we varied the absolute values of $\tau_d$ and $\tau_f$ while keeping $\tau_d \gg \tau_f$ holds ($\tau_f$=0.1, $\tau_d$=3 vs. $\tau_f$=0.3, $\tau_d$=4), and we observed the same statistically significant repulsion effect (both t(19) > 213.97, p < .001), with only its magnitude varying (original: -2.29°; new experiments: -1.34° and -3.61°). We also analyzed the robustness of attraction effect on time constants $\tau_d$ and $\tau_f$ in both single- and two-layer networks, and found that the STP parameter variations do not alter the model performance qualitatively. We will report these results in the revised manuscript.
>
> 2.	About the qualitative comparisons to the experimental results:
> Thanks for raising this question.
> Since our model was built at the neural circuit level, we could not compare neural activities in our model directly to experimental observations. Nevertheless, the performances of our model are consistent with experimental data. These include, 1) our model quantitatively reproduces experimental serial bias amplitudes (\~2°) for both within-trial repulsion and between-trial attraction [Fritsche et al., Curr. Biol. 2017; Cicchini et al. Proc. R. Soc. B 2018; Pascucci et al. PLOS Biol. 2019; Czoschke et al. Br. J. Psychol. 2019; Fischer et al. Nat. Commun. 2020; Moon et al. Psychon. Bull. Rev. 2023]. 2) the judgment error curve as a function of prior-current stimulus difference also closely matches the experimental data. Furthermore, 1) fMRI studies (Schwiedrzik et al., 2014; Sheehan et al., 2022) showed repulsion in early sensory cortices, while attraction involves higher visual-parietal-frontal networks; 2) Luo et al. (2025) presented EEG/MEG evidence of two-stage processing—sensory repulsion and prefrontal attraction. These findings align with the two-layer dynamics of our model: stimulus-specific repulsion via low-layer STD and predictive attraction via high-layer STF. We will add more discussion about these comparisons in the revised manuscript.
>
> 3.	We appreciate the reviewer's comment. We agree that existing evidences only suggest that sensory cortices exhibit STD predominance and higher-level areas show STF predominance, but direct evidence on the task-dependent STD/STF dynamics remains unclear. We will moderate our statement accordingly in the revised manuscript. It is worth mentioning that previous studies noted that the STP timescale aligns with the working memory time window, providing neural substrate for information maintenance [Mongillo et al., Science 2008; Masse et al., Nat. Neurosci. 2019]. Here, our study extends this framework further by demonstrating that the regionally specialized STP critically underlies serial dependence in working memory.
>
> **Questions:**
>
> 1.	About the readout rule of perceptual decision:
> Thanks for this insightful question.
> In our model, perceptual decision was implemented by decoding high-layer network activity via population vector method (PVM). To evaluate potential readout bias from asymmetrical responses, we reanalyzed Fig.3D using three other methods: center-of-mass (COM), maximum likelihood (ML), and peak decoding (Peak). All methods confirm robust within-trial repulsion (serial bias amplitudes/curve peak at: PVM:−0.90°/37.92°; COM:−0.90°/37.92°; ML:−0.97°/36.97°; Peak:−0.61°/45.34°) and between-trial attraction (serial bias amplitudes: PVM:1.17°/19.34°; COM:1.17°/19.34°; ML:1.05°/19.86°; Peak:1.93°/18.21°), which are statistically significant across metrics. We will include these analyses extended to Figs. 2C, 2F, 3E, 3F, and 5 in the revised manuscript.
>
> 2.	Yes, this has been addressed in our response to Question 2 above.
>
> 3.	Thanks for raising this important issue. Our model accounts for most documented serial dependent phenomena and the associated neural mechanisms. We acknowledge that a few experimentally observed patterns require further investigation. Specifically: 1) Attractive biases in primary visual cortex (John-Saaltink et al., 2016) may involve top-down feedback reshaping sensory representations (discussed in Line 310). 2) Long-timescale repulsive effects (>50 s; Fritsche et al., 2020) suggest sensory cortex mechanisms beyond current STP timescales. Implementing long-term negative feedback mechanisms in sensory cortices may explain this. We will expand discussion on these unmodeled studies in the revised manuscript.
>
> 4.	About the possibility for a single-layer network to explain both attraction/repulsion: Thank you for this interesting question. A single-layer network cannot simultaneously account for both attraction and repulsion in sequential dependence due to the antagonistic nature of STD and STF. As Sec. 3 and Fig. 2 demonstrate, repulsion relies on STD-dominance, reducing neural responses and modeling sensory adaptation in primary cortical areas. Conversely, attraction depends on STF-dominance, enhancing responses and supporting information integration in higher-order regions. Since STD and STF modulate synaptic efficacy oppositely, it is not feasible for the same neural population to exhibit antagonistic effects within the same temporal window. Moreover, there is currently no empirical evidence supporting the coexistence of STD-dominant and STF-dominant synapses within the same local circuit. Therefore, explaining bidirectional behavioral effects (attraction and repulsion) necessitates a hierarchical, heterogeneous STP network structure.
>
> 5.	About the importance of STD and STF order: Thank you for raising this important question. The order of STD and STF is crucial for the model performance. This configuration allows the early STD layer to enhance sensitivity to subtle input differences by reducing responses to repeated stimuli, while the downstream STF layer integrates temporal information through response facilitation, thereby stabilizing perception. Reversing this order (STF-dominance first) causes initial signal smoothing due to facilitation. Consequently, even though the downstream STD layer attempts to extract differential features, the loss of sensitivity to input variation cannot be recovered, resulting in failure to achieve the intended synergy between sensitivity and stability. Following your suggestion, we simulated a reversed configuration—lower STF-dominated layer ($\tau_f^L$= 5 s, $\tau_d^L$ = 0.3 s) and higher STD-dominated layer ($\tau_d^H$ = 3 s, $\tau_f^H$ = 0.3 s)—yielded exclusively attractive biases. Both within-trial (amplitude: 1.79°; peaking at 26.79; t(19)=6.79, p<.001; cluster 1°-58°, p<.001) and between-trial effects (amplitude: 3.81°; peaking at 28.07°; t(19)=19.62, p<.001; cluster 1°-87°, p<.001) showed significant attraction.
>
>
> 6.	About the attractor dynamics: We thank the reviewer for raising this insightful question, which deepens our understanding of the role of attractor dynamics. Indeed, attractor dynamics plays a critical role in generating serial dependence effects, driving both within-trial repulsion and between-trial attraction. The stability of neuronal activity states ($r$) depends on synaptic state variables (neurotransmitter release probability $u$, available neurotransmitter resources $x$), with $r$ operating at a faster timescale than $u$ and $x$. Specifically, 1) within-trial repulsion: stimuli create local minima in $x$ state space (Fig. 2AB), suppressing responses to subsequent adjacent stimuli. This forms unstable attractors in $x$-state space, forcing $r$ away from the minima to produce repulsion; 2) between-trial attraction: prior stimuli induce peaks in $u$ state space (Fig. 2DE), enhancing responses to subsequent adjacent stimuli. This establishes stable attractors in $u$-state space, constraining $r$ near specific $u$ values to generate attraction. Notably, the stable attractor mechanism in $u$-state space aligns with Barbosa et al.'s findings (Nat Neurosci 2020) that working memory stores information via activity-silent states maintained by STF-dependent attractors. They reactivate during new trials, biasing ongoing processing.
>
> Finally, we thank the reviewer again for evaluating our manuscript carefully and providing many constructive suggestions, which are very helpful to improve our work. We will incorporate the recommended corrections and additional analyses into the final version of the paper. We hope our responses have addressed all your concerns, and we would be glad to engage in further discussion if needed.

---

> ### Comment · Reviewer_fkF9 · 2025-08-04
> **thanks for your response**
>
> The authors did a good job addressing my concerns. Most of them have been addressed. The new analyses, once incoporated into the paper, should make it a stronger paper. I also read other reviewers’ comments and the authors’ response. In my view, most of the concerns were resolved. Overall, I think this paper should be a nice contribution for NeurIPS.
>
> One of my concerns that were partially (not fully) resolved which may help improve the paper further:
> I agreed with the authors that direct comparsion to the neural data would be challenging and out of scope for the current work, but a more careful/quantitative comparsion to a good behavioral dataset would be useful.  But this is a minor concern given what the authors already showed, and it is up to the authors.

---

> > ### Author Response · Authors · 2025-08-05
> >
> > We thank again the reviewer for the encouraging comments, and the recognition that “this paper should be a nice contribution for NeurIPS”. Yes, we fully agree that a more careful comparison to a good behavior dataset is highly valuable. For instance, we can compare our model performance with the different serial dependence effects elicited by visual stimuli of varying spatial frequencies (low vs. high). The psychophysical experimental data showed that under high spatial frequency condition, serial dependence bias is attenuated (Ceylan, Herzog, & Pascucci, 2021, Cognition). Meanwhile, we also realize that the judgment error curve as a function of prior-current stimulus difference in Fig.3D aligns well with the experimental observations (Fritsche et al., 2017, Curr. Biol., Pascucci et al., 2019, PLOS Biol., Fischer et al., 2020, Nat. Commun.), but we did not discuss about this point in the current manuscript. In the revised manuscript, we will incorporate and discuss about these comparative analyses.

---

### Official Review · Reviewer_kFr8 · 2025-07-06

**Clarity:** 3
**Significance:** 4
**Originality:** 4
**Rating:** 5
**Confidence:** 4

**Summary:**

This paper proposes a two-layer continuous attractor neural network model incorporating synaptic short-term plasticity to elucidate the neural mechanisms underlying repulsive and attractive effect in serial dependence during perception. The lower sensory-processing layer mediates repulsion through neurotransmitter depletion, which drives short-term depression (STD). Conversely, the higher post-perceptionl layer mediates attraction via sustained neurotransmitter release probability, which drives short-term facilitation (STF). The model successfully replicates serial dependence phenomena observed in visual orientation tasks and identifies STP time constants as critical determinants of repulsion/attraction timescales. Furthermore, it offers a biologically implementation for Bayesian interpretations of serial dependence, linking STD to likelihood modulation and STF to prior integration.

**Questions:**

Q1. How does the repulsion induced by STD cross talk with attraction induced by STF between two layers? Does the neurotransmitter depletion influence the accumulation of calcium in upper layer? Are there biological evidences?
Q2.Figure 4 shows the relation with Baysian framework, do we have some mathematical equivalence between two frameworks?
Q3.Figure 5 shows that repulsion or attraction depends on the comparison between time constant $\tau_d,\tau_f$ and ISI/ITI. Are these time constants task dependent? For example, in other perceptual task such as face detection or color discrimination?
Q4.How to distinct the results from STF or top-down cognitive control on the attraction effect?
Q5.Can we extend this model to sequential sound detection further multisensory situation? If so, what should be changed on the model setup?

**Ethical Concerns:**

["NO or VERY MINOR ethics concerns only"]

**Limitations:**

1. STP is uniform in each layer in the model, which is not the case biologically. Author should discuss this simplification on the results.
2.The model ignore the topdown feedback, thus the model can not discuss the effect of attention or working memory on the repulsion or attraction effect, and the goal-oriented bias can not be explained in the present model.
3. There is two-layer continuous attract model on multi sensory integration and Bayesian integration by Wenhao Zhang, the present model should clearly point out the progress of the present model.

**Paper Formatting Concerns:**

The format abides the instructions

**Quality:**

3

**Strengths And Weaknesses:**

Quality：
Strength: 1)The set up of two-layer CANN model is consistent with biological evidence that sensory areas(e.g. V1) often exhibits short term depression while  higher order cortex area(e.g PFC) exhibits short term facilitation.
2) The paper systematically implements the temporal-spatial pattern of repulsion and attraction effect by adjust the parameters such as ISI/ITI and $\Delta$S, which are consistent with classic psychological experiments.
3) The paper connects the STP to the Bayesian framework, STD to likelihood while STP to prior distribution.
Weakness:
1）	The paper needs to screening the effects of parameters one the repulsion and attraction in perception and analyze the robustness of parameters such as $\tau_d$ and $\tau_f$.
2）	The paper did not compare their neural activity with experimental observations such fMRI or electrophysiological recording data.

Stgrength:
1)The logic flow is fluent, starting from the effect of stp in single layer network to reciprocal interaction between two layer and lastly to the Bayesian framework.
2) Figures in the paper can intuitively demonstrate the temporal-spatial dynamics of the network, especially, the prediction of ISI on the transition from repulsion to attraction along with change of ISI in figure 5
Weakness:
Some details of the implementation should be clarified such as the noise term during the simulation.
The biological basis or the reasons to choose some specific value of parameters also should be clarified such $\tau_d$ $\tau_f$ in lower and higher layer of the model.
Provide some explanation or proof on the continuous attractor guaranteed by rotate-transfer invariant.

Significance:
Strength:
Unified the explanation of repulsion and attraction effect using one single model endowed with STP and bridges the gap between Bayesian computation and neural implementation.
Provide some testable experiments, such as ISI on the transition from repulsion to attraction.
Weakness: Lack of the feedback influence from higher order cortex perform attention and decision.

Originality:

Strength:

Clearly proposed that $\tau_d$ and $\tau_f$ as the temporal indication of information integration and separation, which is original.
The paper provide a biological explanation of likelihood and prior in Bayesian computation in neural system.

Weakness:
Authors did not notice the importance of the labor division of STD and STF  in a hierarchical information processing during the  perception.

---

> ### Author Rebuttal · Authors · 2025-07-30
>
> Thanks for your valuable comments and constructive suggestions. Below are our point-by-point responses to your questions.
>
> **Weakness:**
>
> 1. Quality:
>
> Thanks for raising this important question.
>
> (1) Our choice of time constants $\tau_d$ and $\tau_f$ for STD-dominance in sensory cortex and STF-dominance in higher cortex was motivated by the experimental data. To address the concern of the reviewer, we have conducted simulations assessing the robustness of STD/STF dominance in inducing the repulsion/attraction effect. The results showed that while absolute $\tau_d$ and $\tau_f$ values modulate the bias amplitude, it is their relative values determining whether the effect is repulsive or attractive. For instance, varying $\tau_f$ and $\tau_d$ absolute values while maintaining the relationship $\tau_f \gg \tau_d$ (Fig. 2F: $\tau_f$=5, $\tau_d$=0.1; $\tau_d$=4, $\tau_d$=0.3) preserves statistically significant attraction (both t(19) > 85.12, p < .001), although its magnitude varies (from 1.48° in Fig.2F to 1.56° and 1.16°, respectively). Robustness analyses in single- and two-layer networks further confirm that STP parametric variations preserve our findings. These results will be reported in the revised manuscript.
>
> (2) Since our model was built at the neural circuit level, we could not compare neural activities in our model directly to experimental observations. Nevertheless, the performances of our model are consistent with experimental data. These include, for instances, 1) fMRI studies (Schwiedrzik et al., 2014; Sheehan et al., 2022) that show repulsive effects in early sensory cortex, while attraction is associated with higher visual-parietal-frontal networks; 2) EEG/MEG evidence of two-stage processing—sensory repulsion and prefrontal attraction (Luo et al., 2025). These findings align with the two-layer dynamics of our model: stimulus-specific repulsion via low-layer STD and predictive attraction via high-layer STF.
>
> 2. Strength:
>
> Thanks for the valuable comments and suggestions.
>
> (1) In our model, three sources of noises were considered: 1) External Input Noise: included in $I_{ext}(\theta,t)$ (Gaussian white noise, $\mu$: amplitude) during encoding/recall (Line 123); 2) Background Noise: Layer-specific $\mu_{L/H}\xi_b(\theta_{L/H},t)$ in Eqs. 1–2; 3) Synaptic Weight Noise: included in $J_{LL}, J_{HH}, J_{HL}$ (Eqs. 1–2), which were fixed per run but resampled between different runs (each run modelling the behavior data of one participant, Appendix, line 22). Simulations used second-order Runge-Kutta (Eqs. 1–4). Together, these different types of noises account for the variability of within-participant behaviors across trials, and the variability of between-participant behaviors.
>
> (2) Our choice of $\tau_f$ and $\tau_d$ values is motivated by experimental data, which showed that STD-dominance in sensory cortex (Abbott et al., 1997; Chung et al., 2002; Yu et al., 2022) and STF-dominance in PFC (Wang et al., 2006).
>
> (3) Recently, there have been accumulated experimental evidence showing the existence of continuous attractor networks in neural systems. For example, using two-photon imaging on Drosophila's central brain, Kim et al. (Science, 2017) found that heading-direction neurons are aligned in a ring according to their preferred head-directions, forming a ring-like continuous attractor network. The bump of the continuous attractor network continuously shifts along the ring during rotation.
>
> 3. Significance:
>
> Thanks for the insightful comment. Indeed, the current model ignores the top-down feedback from attention/decision-making areas. Notably, even without explicit feedback, our hierarchical model successfully explains serial dependence through layer-specific synaptic plasticity: STD-driven repulsion in the lower circuit versus STF-driven attraction in the higher circuit. This dissociative mechanism may provide a baseline for general serial dependence, upon which attention further modulates the bias in a task-dependent manner. Nevertheless, we will expand discussion (L310–312) on the top-down modulation in shaping serial effects in the revised manuscript. There are experimental studies suggesting that attentional feedback can induce attractive biases even in early visual areas (John-Saaltink et al., 2016), highlighting a key direction for our future research.
>
>
> 4. Originality:
>
> Thank you for recognizing the originality of our work.
> As suggested by the reviewer, in the revised manuscript, we will highlight the importance of the division of STD and STF in hierarchical information processing. Indeed, our model is built upon this biological mechanism: the lower-level circuit is dominated by STD, which provides high sensitivity to input changes, allowing rapid adaptation to current stimuli and the suppression of redundant information. In contrast, the higher-level circuit is dominated by STF, which supports the maintenance of prior information to ensure representational stability, thereby giving rise to the attractive bias.
> This hierarchical dissociation optimizes information processing by functionally decoupling the stability-sensitivity trade-off. Without such separation, STF dominance in early areas would cause excessive feature persistence, impairing subsequent stimulus processing. Similarly, STD dominance in higher areas would disrupt contextual integration. Thus, the layered STD/STF division not only constitutes our core mechanism but aligns with fundamental neural computation principles. We will expand on these optimization perspectives in the Discussion in the revised manuscript.
>
>
> **Questions:**
>
> 1. Thanks for raising this question.
> There is no synapse-level cross-talk between two layers. The cross-talk between two layers in our model is at the neural activity level, which generates different serial dependence biases. Firstly, STD induces repulsive bias in the lower layer. This bias propagates via static feedforward pathways (translation-invariant, non-STP; cf. Yang et al., eLife, 2024; Zhang et al., J. Neurosci., 2016) to the higher layer. Secondly, the bottom-up repulsion competes with the STF-maintained prior information (attractive bias), yielding the observed behavioral pattern—attraction within trials and repulsion between trials.
> This cross-layer competition exhibits time-dependency: psychophysical evidence indicates the attractive effect is strengthened with longer delays (Fritsche et al., 2017), reflecting faster decay of STD-driven repulsion versus STF-sustained attraction. Our model (Fig. 5) captures this time-dependent interaction, where the short timescale of STD leads to a faster decay of repulsion, while STF in the higher layer maintains the attractive bias over longer duration. We will illustrate the dynamics of this crosstalk across layers in the revised manuscript.
>
> 2. We only presented a qualitative link between our model and the Bayesian framework, and there is no formal mathematical equivalence.
>
> 3. We tend to believe that time constants $\tau$s are task-independent, since it is hard for the neural system to change these parameters rapidly in a task-dependent manner. If there are task-dependent serial dependence bias, we tend to believe this is due to some other modulating factors (such as neuromodulators).
>
> 4. Thanks for raising this question. Our model shows that STF-dominance can generate attractive bias, while top-down control may have the similar effect. Since STP is such a general property of neural systems, we tend to believe that the STP-based mechanism is at least involved in serial dependence, with top-down control further modulating the bias flexibly. Anyway, further experimental data is needed to clarify this issue.
>
> 5. Thanks for the interesting question. Our model can be extended to study sequential dependence in the auditory domain, as sound frequency is continuous, auditory cortex neurons exhibit frequency tuning, and the auditory system's topographic and hierarchical organization is preserved throughout the pathway (Wang et al., Annual Review of Neuroscience, 2018; Wang, Nature Neuroscience, 2007). These features align with our model structure.
>
> **Limitations:**
>
> 1. Thanks for pointing out this limitation.
> Actually, we have conducted some preliminary experiments to explore the heterogeneity of STP on the model performance. Specifically, in the STF-dominant network (Fig. 2F), fixing inter-stimulus interval at 1s, we applied ±10% random perturbations to STP parameters of all synapses. Results showed stable attraction amplitude (1.48°) and minimal DoG peak shift (from 24.49° to 24.41°), indicating negligible impact of moderate heterogeneity. Based on this observation, we tend to believe that both the within-trial repulsion (Figure 3D) and the between-trial attraction effect will be sustained under moderate parameter perturbations. In the revised manuscript, we will include these results in Appendix.
>
> 2. We will discuss these limitations as pointed out by the reviewer in the revised manuscript.
>
>
> 3. Thanks for the interesting question. Zhang’s paper studied multisensory integration by using two reciprocally coupled CANNs. Our model also studied information integration, in the form of attraction bias. It will be interesting to explore how the two models can be combined together. For instance, to include STP and sequential stimuli in Zhang’s model, we can explore how serial dependence interact with multisensory integration.
>
> Finally, we thank the reviewer for the insightful comments. Your suggestions are valuable in improving our work, and we are committed to incorporating the revisions and new analyses into the final version. We hope our responses have addressed all your concerns, and we would be happy to engage in further discussion if needed.

---

> > ### Comment · Reviewer_kFr8 · 2025-08-05
> > **Neural Correlates of Serial Dependence: Synaptic Short-term Plasticity Orchestrates Repulsion and Attraction**
> >
> > Thank you, these comments address my concerns. I will maintain my score, which was already 'Accept'.

---

> > > ### Author Response · Authors · 2025-08-05
> > >
> > > Thanks for the support of our work!

---

### Official Review · Reviewer_kncR · 2025-07-16

**Clarity:** 2
**Significance:** 2
**Originality:** 2
**Rating:** 4
**Confidence:** 4

**Summary:**

The authors propose a model endowed with short-term synaptic plasticity to address “serial dependence”. Specifically, the authors aim to reconcile “repulsive” effects (when the current percept is pushed away from a recent sensory input) and “attractive” effects (the percept is drawn toward recent history) within a unified framework. They perform experiments to understand both within-trial and across-trial behavior in their model. Finally, they also provide a loose analogy to Bayesian viewpoints on serial dependence.

**Questions:**

Please refer to my review above.

**Ethical Concerns:**

["NO or VERY MINOR ethics concerns only"]

**Final Justification:**

Interesting modeling effort. Incrementally extends a large body of work in short-term synaptic plasticity, but nonetheless relevant to the NeurIPS community.

**Quality:**

2

**Strengths And Weaknesses:**

This paper presents a concise and incremental contribution to the literature on short-term synaptic plasticity in neural network models. They tackle an interesting question – serial dependence. I do believe that this would be a useful addition to the literature. However, I’m concerned that the current manuscript appears preliminary and requires substantial revision before it is ready for publication. I’ve detailed some of the concerns/comments below.

I like the start of the introduction. It sets up a clear phenomenon. However, the flow is interrupted by the “related works” section before returning to the introduction. My suggestion to the authors is to have a dedicated “Contributions” subsection at the end of the intro (and before the related works section). This will allow them to complete stating the question, their approach, and what they bring to the table.

Figure 2 needs significant improvement, particularly in terms of captioning and legends.

- I don’t understand what the red horizontal lines are in A, D

- I’m guessing that the different saturated lines in C, F are different ISIs? I had initially thought this was the length of the delay period but it was mentioned otherwise somewhere else in the text. What do the shaded lines represent? Are these standard errors across runs?
Why are the authors showing $x$ in A and $u$ in B? This wasn't very clear at first. Why not show both $x$ and $u$ for both the panels? Or $x \times u$

Overall, the magnitude of the effects reported in Fig. 2 appears very marginal. This is not a compelling demonstration. Did the authors perform any statistical tests to support their findings? Also, is the magnitude of effects reported in prior psychophysics literature consistent with what’s being described here (i.e., a maximum of ~2 degrees either way)? This seems very important to clarify.

Additionally, my understanding is that even this effect completely relies on the choice of $\tau$s. I would like to see some kind of sensitivity analysis to understand how robust any of these effects are. It’s possible that the authors chose the values for these time constants based on biological (not modeling) literature. If so, that would also be additionally useful to clarify.

One suggestion to sort of bypass this comment about hyperparameter choices is that the $\tau$s can be learnable and the model is trained (with gradient descent) to report the perceived orientation. For example: Fig. 4 in Linsley et al. ICLR 2020. Recurrent Neural Circuits For Contour Detection. This is not exactly the same kind of thing the authors are going for, but I’m mentioning this as a motivating example.

It also seems to me that it’s important to show negative ISIs. The authors mention that “without loss of generality,” they always cue the second stimulus, but it is not clear to me why this is in principle the same as cueing the first one. In one case, the memory of the stimulus seems way more important. Can the authors clarify?

The parallels to the Bayesian framework, though interesting, are very vague and unclear to me. Are the authors arguing that, depending on the magnitude of instantaneous noise, the network’s readout is non-deterministic? I didn’t find any experiments systematically testing the effect of noise in this context.

L147 “modeling behaviors of different participants”. What does this mean?

Some grammatical errors that I could spot. Please do a thorough proofread.
L122 “cuing”
Fig. 2 Caption “The detailed simulations, see Appendix A”
L153 “manupilating”

---

> ### Author Rebuttal · Authors · 2025-07-30
>
> Thanks for the valuable comments. We sincerely appreciate the detailed and constructive suggestions. Below, we provide a point-by-point response to each of the questions.
>
> 1.	According to your suggestion, revised manuscript adds 'Contributions' subsection to Introduction end and moves 'Related Works' post-Introduction.
>
> 2.	Thanks for pointing out the unclear descriptions in Fig.2. We will clarify them in the revised manuscript.
>
> (1)	In Fig.2A,D (top), the red line marks the decoded stimulus orientation. In the bottom panel, the red lines during stimulus/delay periods denote stimulus orientations in current trial (i.e., $\theta_1^s=-30\degree, \theta_2^s=0\degree$). The recall-period red line indicates the recalled orientation of the second stimulus ($\theta_2^d$).
>
> (2)	Yes, Fig. 2C, F demonstrate adjustment errors under different ISI conditions. The shaded area represents standard error across simulation runs.
>
> (3)	In Fig.2A, we choosing to only illustrate the dynamics of the available neurotransmitter $x$ is to highlight that in the STD-dominated condition ($\tau_d \gg \tau_f$), the slow dynamics of $x$ leads to the repulsion effect; while the neurotransmitter release probability $u$ rapidly returns to the baseline and has little contribution. We have the same consideration for only illustrating the dynamics of $u$ in Fig.2D. The reason for we not illustrating the dynamics of $x \times u$ is that it confounds the dynamics of $x$ and $u$ in the STD- or STF-dominated condition. But, as concerned by the reviewer, we will add illustrations of the dynamics of $u$, $x$, and $x \times u$ in Appendix in the revised manuscript.
>
> 3.	Thanks for raising this important question. Please note that a serial bias of \~2 degree is commonly observed in psychophysical experiments (Fritsche et al., Current Biology, 2017; Pascucci et al., PLOS Biology, 2019; Czoschke et al., British Journal of Psychology, 2019; Fischer et al., Nat Commun, 2020; Moon et al., Psychonomic Bulletin & Review, 2023). Our model performances are highly consistent with psychophysical experimental observations, which include two parts (Fig.3D-F): 1) in the post-cueing task, our model produces the serial bias amplitudes agreeing with the experimental data (\~2 degree); 2) in our model, the judgement error curve against the difference between the prior and current stimuli also agrees well with the experimental data.
>
> As concerned by the reviewer, we performed statistical tests on all relevant model performances, which are all statistically significant. We will include these results in the revised manuscript. As an example, here we present the statistical tests for Fig.2C,F. Specifically, we conducted one-sample t tests and cluster-based permutation tests. One-sample t test revealed significant repulsion effect induced by STD-dominance in all ISI conditions (all t(19) > 43.18, p < .001; Fig. 2C) and significant attraction effect induced by STF-dominance in all conditions (all t(19) > 85.63, p < .001; Fig. 2F). These effects were confirmed by using a cluster-based permutation test, showing a significant cluster (p < .001) when $\Delta S$ was between 1° and 88°.
>
> 4.	Thanks for raising this important issue. Indeed, our choice of time constants $\tau$s for STD-dominance and STF-dominance is based on the biological data (see e.g, Abbot et al., Science, 1997; Chung et al. Neuron, 2002; Yu et al., Neuron, 2022; Wang et al., Nat. Neurosci., 2006). But as pointed out by the reviewer, since these parameters are so critical, it is worth analyzing the sensitivity of model on their values, which at least justifies why the biological system adapts to this parameter regime. We therefore conducted the sensitivity analyses. For example, for the STD-dominance condition (Fig.2C), we varied the absolute values of $\tau_d$ and $\tau_f$ while keeping $\tau_d \gg \tau_f$ holds ($\tau_f$=0.1, $\tau_d$=3 vs. $\tau_f$=0.3, $\tau_d$=4), we observed the same statistically significant repulsion effect (both t(19) > 213.97, p < .001), with only its magnitude varying (original: -2.29°; new experiments: -1.34° and -3.61°). We also analyzed the sensitivity of the attraction effect on time constants $\tau$s in both single- and two-layer networks, found that the STP parameter variations do not alter the model performance. We will report these results in the revised manuscript.
>
> 5.	Thanks for raising this question. We tend to believe that time constants $\tau$s are invariant when the brain is performing a cognitive task, since it is hard for the neural system to change all neurons’ $\tau$s rapidly. If there exists task-dependent serial dependence effect, we tend to believe this is due to some modulating factors (such as neuromodulators). Nevertheless, it is interesting to train a recurrent neural network with learnable $\tau$s and a properly designed loss function to explore serial dependence effects, and if the trained network does not exhibit the association between biases and STP-dominance as proposed in this work, it will further strengthen our model.
>
> 6.	Thank you for raising this issue. We apologize for the confusion in our descriptions. Actually, except in the one-layer model where we cued the second stimulus only (Fig.2), we always cued both stimuli in the two-layer model. Specifically, in the two-layer model experiments (Fig. 3–5), we employed a recall paradigm where either S1 or S2 was randomly cued in each trial, as described on L183 in the manuscript. Based on this paradigm, we computed the within/between-trial serial dependence effects, and our results are consistent with experimental findings. In the one-layer model (Fig. 2), we did cue the 2nd stimulus only. To address your concern – whether cueing S2 was the same as cueing S1, we conducted additional analyses. We re-run the one-layer model by randomly cueing S1 or S2 in interleaved trials. The results show that in STD-dominance, the adjustment error curve aligns with that in Fig.2C, with a magnitude of -3.51° (t(19) = 168.57, p < .001); and in STF-dominance, the adjustment error curve aligns with that in Fig. 2F, with a magnitude of 1.68° (t(19) = 96.97, p < .001). These results indicate that the choice of recalled stimulus (S1 or S2) does not affect our conclusion.
>
> We will clarify the above confusion in the revised manuscript and add the new analysis results.
>
> 7.	Thanks for raising this important issue. The serial dependence effect is indeed not deterministic but a statistical phenomenon over many trials. However, here we linking our model to the Bayesian framework is for another purpose, that is, we want to show that our model provides a potential neural mechanism for the widely recognized Bayesian interpretation of serial dependence in the field (Pascucci et al., Plos Biology, 2019). Let us explain this in more detail.
>
> The Bayesian framework for serial dependence consists of two stages:
>
> (1)	Sensory Encoding Stage: The Bayesian framework posits that, after exposure to the previous stimulus ($\theta_n$, Fig. 4A bottom), neurons preferring stimuli near $\theta_n$ undergo adaptation, manifested as reduced responses to new inputs. This alters the likelihood function in the Bayesian framework and generates the repulsion effect in sensory perception.
>
> In the first layer of our model, due to STD-dominance, the available neurotransmitter $x$ decays after the presence of the first stimulus, and the distribution of $x$ values across the neuron population is non-uniform (black curve in Fig. 4B bottom), with more active neurons having smaller $x$. When the second stimulus is presented, this non-uniform distribution of $x$ across the neuron population induces alternated responses compared to that to the first stimulus, which can be regarded as the likelihood function is changed in the Bayesian framework.
>
> (2)	Post-Perception Decision Stage: The Bayesian framework posits that higher cortical areas develop weight bias toward the prior stimulus ($\theta_n$, Fig. 4A top), modifying the prior distribution and inducing the attraction effect.
>
> In the second layer of our model, due to STF-dominance, the neurotransmitter release probability $u$ (green curve, Fig. 4B middle) increases at the location of the previous stimulus ($\theta_{cue}^{d,n}$), which facilitates neuronal responses at the previous stimulus location, as if the prior distribution of stimulus is changed in the Bayesian framework.
>
>    In summary, our model provides a qualitative interpretation of the Bayesian framework for serial dependence, and we hope this link helps us to better understand the neural mechanism of serial dependence.
>
> 8.	Thanks for raising this question. In our model, we used each run with slightly different model parameters to represent the behavior of each simulated individual, with 20 runs representing the behavioral data of 20 participants. Specifically, to model the variability between participants, we incorporated noises into the synaptic connections within and between layers ($J_{LL}, J_{HH}, J_{HL}$ in Eqs.1-2), and these noises were independently sampled in each run, leading to slightly different network responses (see Appendix, lines 22–24). Furthermore, to model the trial-to-trial variability within-participant, we incorporated noises into the input signal across 100 trials within each run. The above two types of noises simulate, respectively, the inter-individual and intra-individual variabilities in experiments.
>
> 9.	Thanks for pointing out grammar errors. We will thoroughly proofread the revised manuscript.
>
> Finally, we thank the reviewer for carefully reviewing our manuscript and providing many constructive suggestions, which are highly valuable to improve our work, and we commit to incorporating the corrections and new analyses into the final version of the paper. We hope our replies have addressed all concerns of the reviewer, and if not, please feel free to discuss with us.

---

> > ### Comment · Reviewer_kncR · 2025-08-05
> > **acknowledging the rebuttal**
> >
> > Dear authors: Thank you very much for your detailed response. I appreciate the answers and the supporting analyses. I have now gone through your response to my comments as well as the comments from other reviewers. Most of my questions have been answered, particularly related to the magnitude of the effects.
> >
> > The contribution here is incremental, but still interesting to the NeurIPS community. I am increasing my score to a 4. Good luck to the authors.

---

> > > ### Author Response · Authors · 2025-08-06
> > >
> > > We sincerely appreciate again the reviewer for the encouraging comments and raising the score.

---

### Note · Authors · 2025-08-13

We sincerely thank all reviewers for their valuable comments and encouraging feedback on this work. To better assist the Area Chair's evaluation, we would like to summarize the key contributions and strengths of our work, as recognized by reviewers, which are:

1. The research motivation is clear, the topic is novel, and the logic is rigorous; we proposed a neural mechanism model that can unify the explanation of both attractive and repulsive serial dependence effects in perception (Reviewer kncR, kFr8, fkF9, rAyR).
2. The model results are consistent with behavioral data across a variety of experimental conditions (Reviewer kncR, kFr8, fkF9, rAyR).
3. The model is dominated by short-term depression (STD) at lower levels and by short-term facilitation (STF) at higher levels, which aligns with existing neurophysiological evidence and provides a potential neural basis for Bayesian inference (Reviewer kncR, kFr8, fkF9, rAyR).
4. The model offers testable predictions for future experiments (Reviewer fkF9).

During the rebuttal, we addressed the concerns raised by reviewers with the following responses and additional analyses:

1. Model parameter selection: The time constants were determined based on biological data. We also conducted sensitivity analyses, showing that variations in the time constants do not alter the main conclusions of the model (Reviewer kncR, kFr8, fkF9).
2. Comparison with experimental data: The amplitudes of the attractive and repulsive curves predicted by the model (~2 degrees) match experimental results; we will add a direct comparison between the two. We also performed significance tests, which confirmed that both attractive and repulsive effects are significant (Reviewer kncR, fkF9).
3. Comparison with Bayesian accounts: We further clarified the relationship between our model and the Bayesian framework (Reviewer kncR, rAyR).

During the discussion, we were pleased to see that two reviewers stated that all of their concerns were resolved, and the other two reviewers were satisfied that their major concerns were addressed and expressed that our paper is suitable for acceptance.

Finally, we once again thank all reviewers for their thorough evaluations and constructive feedback. We are committed to incorporating reviewers' insightful suggestions, along with extra analyses and clarifications, into the final version of the manuscript.

---

### Decision · Program_Chairs · 2025-09-17

**Decision:**

Accept (poster)

**Comment:**

This paper studies our understanding of how the neural system leverages synaptic short-term plasticity to balance sensitivity in sensory perception with stability in post-perceptual cognition using a two-layer continuous attractor neural network. The reviewers found the paper well written on an important topic for the NeurIPS community. The rebuttal was engaging and addressed most of the reviewers’ concerns. All four reviewers unanimously recommended acceptance. Addressing the concerns raised in the reviews and including the rebuttal discussions in the final version will improve the paper quality.